# Bootstrap3D: Improving Multi-view Diffusion Model with Synthetic Data

## Abstract

Recent years have witnessed remarkable progress in multi-view diffusion models for 3D content creation. However, there remains a significant gap in image quality and prompt-following ability compared to 2D diffusion models. A critical bottleneck is the scarcity of high-quality 3D data with detailed captions. To address this challenge, we propose **Bootstrap3D**, a novel framework that automatically generates an arbitrary quantity of multi-view images to assist in training multi-view diffusion models. Specifically, we introduce a data generation pipeline that employs (1) 2D and video diffusion models to generate multi-view images based on constructed text prompts, and (2) our fine-tuned 3D-aware **MV-LLaVA** for filtering high-quality data and rewriting inaccurate captions. Leveraging this pipeline, we have generated 1 million high-quality synthetic multi-view images with dense descriptive captions to address the shortage of high-quality 3D data. Furthermore, we present a **Training Timestep Reschedule (TTR)** strategy that leverages the denoising process to learn multi-view consistency while maintaining the original 2D diffusion prior. Extensive experiments demonstrate that Bootstrap3D can generate high-quality multi-view images with superior aesthetic quality, image-text alignment, and maintained view consistency.

## 1 Introduction

3D content creation stands as a fundamental challenge within the generative domain, boasting widespread applications in augmented reality (AR) and game modeling. Unlike 2D image generation, the dearth of high-quality 3D models persists as a significant hurdle to overcome. In the realm of 2D image generation, the pivotal role of training on billion-scale image-text pairs (Schuhmann et al., 2022) has been firmly established (Betker et al., 2023; Rombach et al., 2022; Li et al., 2024; Chen et al., 2023a; 2024a). However, in 3D content generation, the scarcity of high-quality 3D models often compels reliance on the priors of 2D diffusion models. The predominant methodologies in this domain can be categorized into two main streams: 1) Gaining optimized neural representations from fixed 2D diffusion models via Score Distillation Sampling (SDS) loss (Poole et al., 2022; Shi et al., 2023b; Liu et al., 2023b; Shi et al., 2023a; Liu et al., 2023a; Wang et al., 2024a), which are time-intensive, lacking diversity and suffer from low robustness although capable of producing high-quality 3D objects. 2) Fine-tuning 2D diffusion models to achieve multi-view generation (Li et al., 2023a; Shi et al., 2023a;b) , directly synthesizing 3D objects through sparse reconstruction models (Li et al., 2023a; Wang et al., 2023b; Xu et al., 2024a;b; Tang et al., 2024a; Wei et al., 2024). With recent improvements in large-scale sparse view reconstruction models and 3D representations (Kerbl et al., 2023), the second stream is garnering increasing attention.

Fine-tuning 2D diffusion models for multi-view generation remains challenging owing to the insufficiency in both data quality and quantity. Previous methods (Qiu et al., 2023; Li et al., 2023a; Shi et al., 2023b; Deitke et al., 2024) primarily train on a filtered subset of high-quality data from Objaverse (Deitke et al., 2023) and Objaverse-XL (Deitke et al., 2024). The scarcity of high-quality data often introduces various shortcomings. In single-view based novel view synthesis (Liu et al., 2023b; Shi et al., 2023a; Wang & Shi, 2023; Voleti et al., 2024), if the input images deviate from the distribution of the training data, it can induce issues such as motion blurring, object distortion and deformation (Shi et al., 2023a).

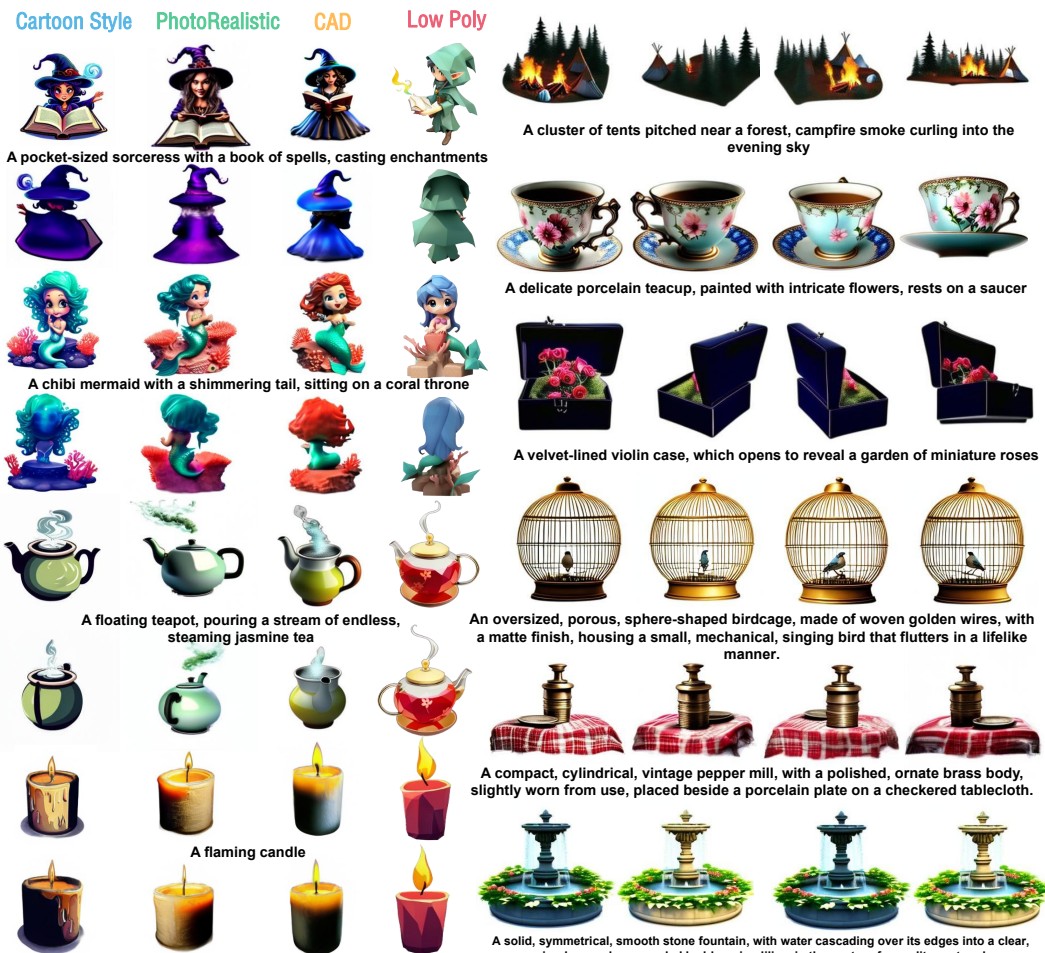

Figure 1: **Bootstrap3D** can generate high quality multi-view images with precise long text control and style customization while maintaining view consistency.

Moreover, in direct text-to-multi-view image generation, the pursuit of enhancing view consistency compromises the aesthetic and photo-realistic quality. For instance, Intant3D (Li et al., 2023a) fine-tunes SDXL (Podell et al., 2023) using only 10K high-quality Objaverse (Deitke et al., 2023) data with a small learning rate for 10K steps, which does not fundamentally prevent the catastrophic forgetting problem of losing 2D diffusion prior, leading to compromised image quality. Recent endeavors have predominantly focused on alleviating data scarcity and improving view consistency from a model-centric perspective (Kant et al., 2024; Shi et al., 2023a; Tang et al., 2024b), with limited exploration into the improvement of training data and training method itself.

Recent Multimodal Large Language Models (MLLMs) (Liu et al., 2024a; Chen et al., 2023b; Li et al., 2023b; Alayrac et al., 2022; Anil et al., 2023) like GPT-4V (OpenAI, 2023a) and Gemini (Team et al., 2023), possess image understanding capabilities and rudimentary 3D world aware-ness, has enabled automatic quality assessment of multi-view images and dense caption generation. Furthermore, notable advancements in video diffusion (Brooks et al., 2024; Voleti et al., 2024) have improved the generalizability of novel view synthesis (Voleti et al., 2024; Chen et al., 2024b; Kwak et al., 2023). Employing these advancements, we propose Bootstrap3D to generate synthetic data to counteract the data deficiencies inherent in training multi-view diffusion models. To be specific, we introduce the Bootstrap3D data generation pipeline for producing high-quality multi-view im-ages with dense descriptive captions. Subsequently, we fine-tune a multi-view-aware MLLM model, dubbed as MV-LLaVA, to achieve fully automated high-quality data annotation with both efficiency and accuracy. To mitigate catastrophic forgetting of 2D diffusion prior, we introduce a training timestep reschedule (TTR) strategy when fine-tuning multi-view diffusion models. Specifically, we use the phased nature of the denoising process (Ho et al., 2020) and limit different training time steps for synthetic data to achieve enhanced image quality with maintained view consistency.

Through extensive experiments, we demonstrate that our method significantly enhances the adherence of the multi-view diffusion model to text prompts and image quality while ensuring view consistency. Integrated with the reconstruction model, our approach facilitates the creation of 3D models with superior quality. We show some of the qualitative results in Fig. 1, where our model can achieve high quality multi-view images with precise text control and style customization. Our contributions are summarized into the following points:

**1)** We present an automated **Bootstrap3D** data generation pipeline that uses the video diffusion model and our fine-tuned 3D-aware MV-LLaVA to synthesize an arbitrary number of high-quality multi-view image text pairs.

**2)** We propose a Training Time-step Reschedule (TTR) strategy for fine-tuning the multi-view diffusion model that employs both synthetic data and real data to enhance image quality and image-text alignment while maintaining view consistency.

**3)** We generate 1 million multi-view images with dense descriptive captions suitable for training the multi-view diffusion model and provide dense descriptive captions on Objaverse Deitke et al. (2023), which mitigates the gap with the 2D diffusion model from a data perspective.

## 2 RELATED WORK

**Existing 3D datasets and data pre-processing.** Existing object level 3D datasets, sourced either from CAD (Chang et al., 2015; Wu et al., 2015; Deitke et al., 2023; 2024) or scan from real objects (Aanæs et al., 2016; Yao et al., 2020; Downs et al., 2022; Wu et al., 2023), are still small in size. Most state-of-the-art open-sourced 3D content creation models are trained on Objaverse (Deitke et al., 2023). However, there still exists a huge gap compared to data used for training 2D diffusion models (Schuhmann et al., 2022). In addition to quantity, quality is also an important problem remains to be solved as many methods (Shi et al., 2023b; Li et al., 2023a; Qiu et al., 2023; Tang et al., 2024a) trained on Objaverse rely on filtering out low-quality data, making the precious 3D data even less. Another critical gap that requires attention is the quality of the 3D object's caption. Previous work Cap3D (Luo et al., 2024) propose to apply BLIP-2 (Li et al., 2023b) and GPT-4 (OpenAI, 2023b) to generate caption based on multi-view images. However, this approach, without direct input image into GPT, can lead to severe hallucination. Given recent breakthroughs in improving text-image alignment through caption rewriting (Betker et al., 2023; Chen et al., 2023a; 2024a; Esser et al., 2024), there is a pressing need to rewrite denser and more accurate captions for 3D objects with the assistance of advanced Multimodal Large Language Models (MLLMs). In this work, we propose a new data generation pipeline to synthesize multi-view images and rewrite captions for 3D objects incorporating additional quality scoring mechanisms to address the aforementioned issues.

**Text-to-3D content creation.** The field of 3D content creation has been a vibrant area of research over the past years. One prominent research direction explores the use of Score Distillation Sampling (SDS) (Poole et al., 2022) and its variants (Chen et al., 2023c; Chung et al., 2023; Hertz et al., 2023; Liang et al., 2023; Lin et al., 2023; Liu et al., 2023b; Shi et al., 2023b; Liu et al., 2023c; Long et al., 2023; Wang et al., 2024a; Tang et al., 2023; Wang et al., 2023a; Yang et al., 2024; Qi et al., 2024), using the priors of 2D diffusion models to optimize 3D representations. While these methods have demonstrated success in producing high-quality 3D generations, they often require prolonged optimization time to converge. In contrast, recent studies (Hong et al., 2023; Wang et al., 2023b; Li et al., 2023a; Tang et al., 2024a; Tochilkin et al., 2024; Xu et al., 2024b; Wei et al., 2024) have proposed the direct inference of 3D representations (Mildenhall et al., 2021; Chan et al., 2022; Kerbl et al., 2023; Zhang et al., 2023a) conditioned by images. Among these approaches, Instant3D (Li et al., 2023a) stands out by utilizing multi-view images of the same object to directly deduce the Triplane (Chan et al., 2022) representation. This approach effectively addresses the issue of ambiguous unseen areas inherent in the single image to 3D conversions, as encountered in LRM (Hong et al., 2023) and TripoSR (Tochilkin et al., 2024). Instant3D, along with subsequent works (Xu et al., 2024b; Zheng et al., 2024; Wang et al., 2024b; Xu et al., 2024a), efficiently decomposes 3D generation into two processes: the generation of multi-view images using multi-view diffusion model (Liu et al., 2023b;c;a; Shi et al., 2023b; Liu et al., 2024b; Shi et al., 2023a; Long et al., 2023; Kant et al., 2024; Voleti et al., 2024) and large reconstruction model to generate 3D representations conditioned on these multi-view images. In this work, we introduce a method that significantly enhances the scalability of training and data generation for multi-view image generation.

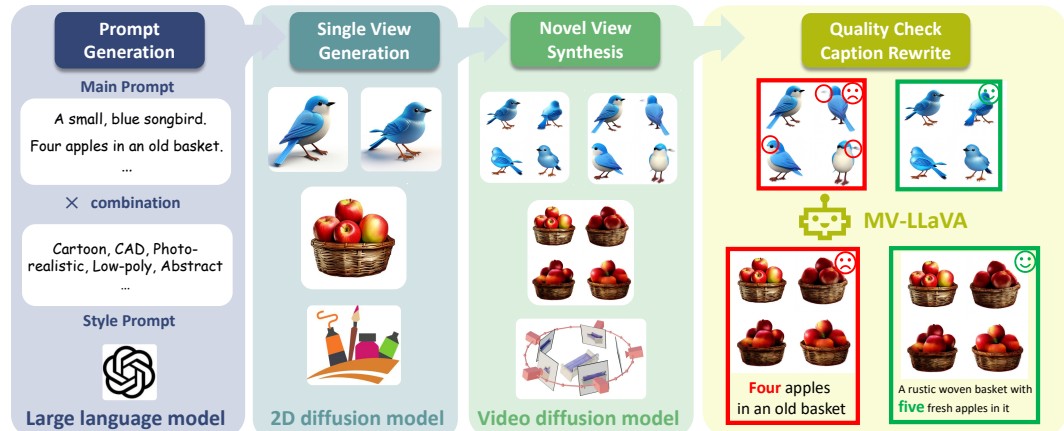

Figure 2: **Bootstrap3D data generation pipeline** that consists of 1) using LLM to generate diverse text prompts 2) employing the T2I model to generate single-view images 3) synthesizing arbitrary number of multi-view images by applying the video diffusion model, 4) employing MV-LLaVA to filter and select only high-quality data, and rewrite captions to be dense and descriptive.

**Video diffusion for novel view synthesis.** Recent advancements in video diffusion have marked a significant breakthrough, with models such as Sora (Brooks et al., 2024) and SVD (Blattmann et al., 2023) scaling up the direct generation process from images to videos. Following these developments, a series of works (Wang et al., 2023c; Kwak et al., 2023; Blattmann et al., 2023; Melas-Kyriazi et al., 2024; Han et al., 2024; Chen et al., 2024b) represented by SV3D (Voleti et al., 2024), have fine-tuned these video diffusion models for 3D content creation. Despite these groundbreaking developments, the new perspective images generated based on video priors still suffer from issues like motion blur. In this work, we propose to utilize SV3D (Voleti et al., 2024) as a data generator to produce novel views of given images with additional quality checks to leave only high-quality data.

**Multimodal Large Language Models.** With the development of large language models (Brown et al., 2020; OpenAI, 2023b; Chowdhery et al., 2022; Anil et al., 2023; Hoffmann et al., 2022; Touvron et al., 2023), multimodal large language models (MLLMs) (Zhang et al., 2023b; Alayrac et al., 2022; Li et al., 2023b; 2022; Huang et al., 2023; Driess et al., 2023; Awadalla et al., 2023; Liu et al., 2024a; Dong et al., 2024; Sun et al., 2023), such as GPT-4V (OpenAI, 2023a), have demonstrated groundbreaking 2D comprehension capabilities and open-world knowledge. As is discovered in GPTEval3D (Wu et al., 2024), GPT-4V can achieve human-aligned evaluation for multi-view images rendered from 3D objects. In this work, we fine-tune the LLaVA (Liu et al., 2024a) for quality judgment and descriptive caption generation based on multi-view images.

## 3 METHODS

Due to the scarcity of high-quality 3D data, we develop the Bootstrap3D data generation pipeline to efficiently construct an arbitrary number of training data (Sec. 3.1). Subsequently, the quality of generated multi-view images is assessed using the powerful GPT-4V (OpenAI, 2023a) or our proposed MV-LLaVA (Liu et al., 2024a) model to generate dense descriptive captions efficiency and faithfully (Sec. 3.2). We also design a training timestep reschedule (Sec. 3.3) when fine-tuning the multi-view diffusion model with our synthetic and real data.

### 3.1 BOOTSTRAP3D DATA GENERATION PIPELINE

As illustrated in Fig.2, our data generation pipeline initially employs GPT-4 (OpenAI, 2023a) to generate a multitude of imaginative and varied text prompts (Wu et al., 2024). Subsequently, to generate 2D images that closely align with the text prompts, we utilize the PixArt-Alpha (Chen et al., 2023a) model use FlanT5 (Chung et al., 2024) text encoder with DiT (Peebles & Xie, 2023) architecture for text-to-image (T2I) generation. Thereafter, we use SV3D (Voleti et al., 2024) for novel view synthesis. Given the significant motion blur and distortion often present in SV3D (Voleti et al., 2024) outputs, we further employ Multimodal Large Language Models(MLLM) to evaluate the quality of multi-view images. To rectify mismatches between multi-view images and the original

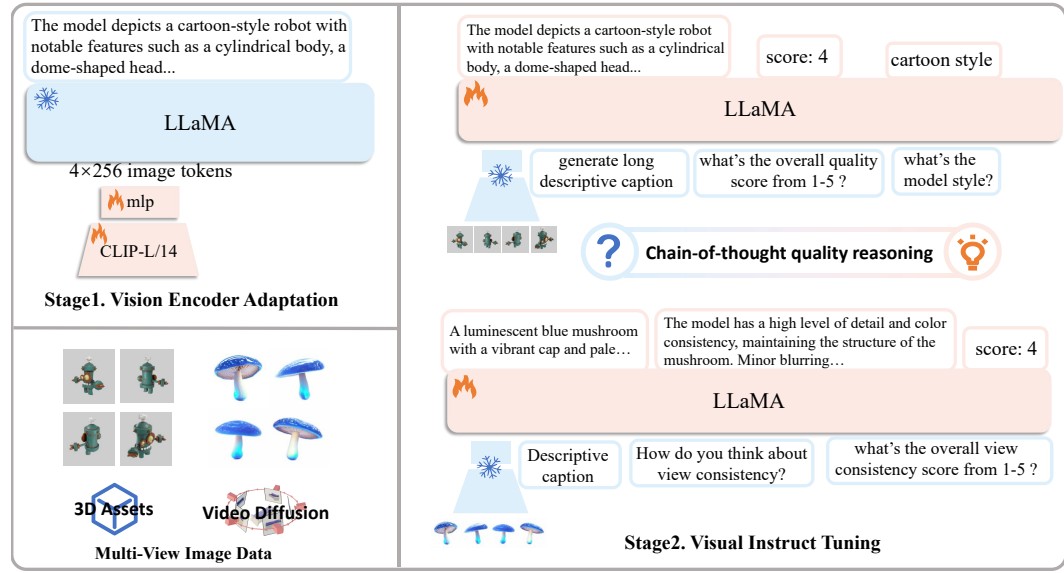

Figure 3: **MV-LLaVA.** We use GPT-4V (OpenAI, 2023a) to generate long descriptive captions, quality scoring, and reasoning processes for multi-view images to construct the instruction tuning dataset. Then we fine-tune our MV-LLaVA based on LLaVA (Liu et al., 2024a) to serve as the human-aligned quality checker and captioner for multi-view images.

text prompts induced by novel view synthesis and provide more precise captions, we further propose MV-LLaVA to generate dense descriptive captions for multi-view images.

## 3.2 MULTI-VIEW LLAVA (MV-LLAVA)

To efficiently generate captions and label quality scores for both generated multi-view images and 3D assets in Objaverse (Deitke et al., 2023), we propose the Multi-View LLaVA (MV-LLaVA) that fine-tune LLaVA (Liu et al., 2024a) based on our instructive conversation pairs generated by the powerful GPT-4V (OpenAI, 2023a).

**Preparing the instruction tuning data.** As shown in Fig.2, we use GPT-4 to generate 20k varied text prompts based on prompts designed in (Wu et al., 2024) and use PixArt-alpha (Chen et al., 2023a) to generate single view image and use SV3D (Voleti et al., 2024) or Zero123++ (Shi et al., 2023a) to generate multi-view images. For these 20k generated multi-view images, we prompt GPT-4V (OpenAI, 2023a) to generate comments on view consistency, image quality and generate dense descriptive captions. For the additional 10K rendered multi-view images from Objaverse (Deitke et al., 2023), we prompt GPT-4V (detailed prompts in Sup. A.5.1) to offer feedback on the quality and aesthetic appeal of 3D objects, along with style judgments. We utilize these 30K high-quality multi-view image text pairs (prompts detailed in Sup. A.5.2) as the instruction tuning data for LLaVA.

**Instruction tuning.** As presented in the left part of Fig. 3, due to the LLaVA's maximum training context length constraints of 2048, we input four images separately into CLIP-L/14 (Radford et al., 2021) and generate 4×256 image tokens. Inspired by ShareGPT-4V (Chen et al., 2023b), we freeze only a portion of layers of CLIP (Radford et al., 2021) in the first stage of pre-training to enhance multi-view awareness and texture perception of vision encoder (detailed in Sup. A.4.1). As shown in the right part of Fig. 3, we first ask the model to generate descriptions, then let the model score the quality based on multi-view images and captions. Our approach encourages LLM to deduct more reasonable quality scores through chain-of-thought (Wei et al., 2022). A mixture of original training data of LLaVA is adopted to mitigate over-fitting. As a result, we obtain MV-LLaVA, which efficiently filters and re-captions both synthetic data and 3D assets. As detailed in Sup.A.4, MV-LLaVA can not only generate more accurate, less hallucinated dense captions that faithfully describe 3D objects compared to Cap3D (Luo et al., 2024) but also assign the human-aligned quality score on both synthetic data and Objaverse assets. The filtered high-quality multi-view images with rewritten dense captions served as training data for the diffusion model.

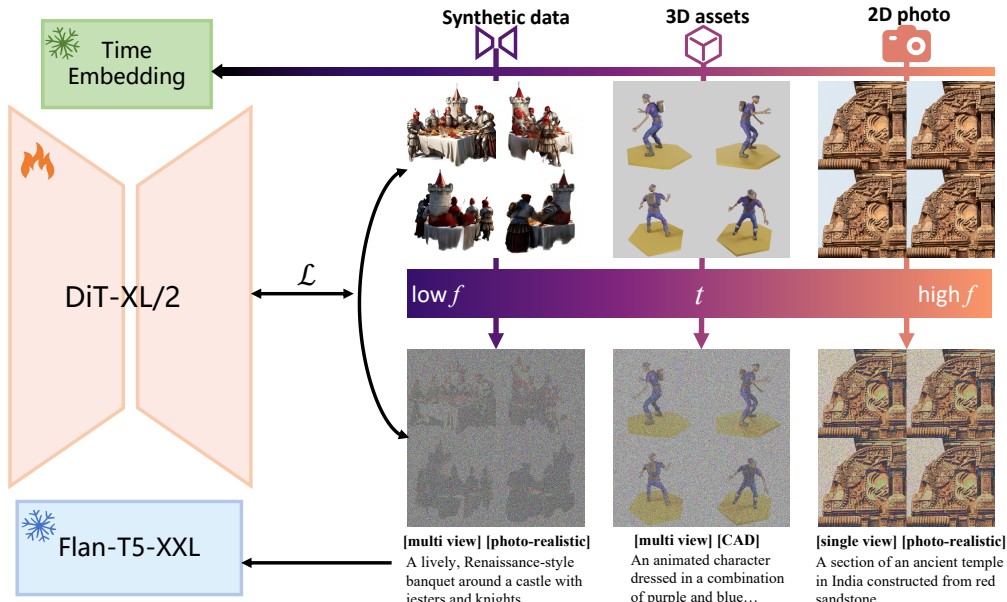

Figure 4: **Training Timestep Reschedule (TTR).** For different types of training data, we restrict the training time step $t$ accordingly to achieve the balance between varied high aesthetic images that are better aligned with text prompt, photo-realistic texture, and view consistency for 3D generation.

### 3.3 TRAINING TIMESTEP RESCHEDULE (TTR)

Despite retaining only relatively high-quality synthetic data with minimal motion blur from SV3D (Voleti et al., 2024) through MV-LLaVA, small areas of blurring persist, stemming from both motion and out-of-distribution scenarios for SV3D and SVD (Blattmann et al., 2023). These blurred data can potentially compromise the final performance of the multi-view diffusion model. To restrict the training time step for synthetic data, we proposed a simple yet effective Training Timestep Reschedule (TTR) method.

**Background.** Before delving into TTR, we briefly review some basic concepts needed to understand diffusion models (DDPMs) (Ho et al., 2020; Sohl-Dickstein et al., 2015; Salimans & Ho, 2022; Rombach et al., 2022; Chen et al., 2023a). Gaussian diffusion models assume a forward noising process which gradually applies noise to real data $x_0$

$$q(x_t|x_0) = \mathcal{N}(x_t; \sqrt{\bar{\alpha}_t}x_0, (1 - \bar{\alpha}_t)\mathbf{I}) \tag{1}$$

here constants $\bar{\alpha}_t$ are hyperparameters. By applying the reparameterization trick, we can sample

$$x_t = \sqrt{\bar{\alpha}_t}x_0 + \sqrt{1 - \bar{\alpha}_t}\epsilon_t \tag{2}$$

During training, $t$ is randomly sampled in $[0, N]$ ($N = 1000$ in (Chen et al., 2023a; Rombach et al., 2022)) for the model to predict the added noise $\epsilon_t$, where $x_0$ denotes for the clear nature image and $x_N$ denotes for pure Gaussian noise. As depicted in Fig.4, when $t$ is large, the denoising process primarily focuses on determining the global low frequency($f$) content such as overall structure and shape. Conversely, when $t$ is small, the denoising process is predominantly responsible for generating high $f$ components such as texture.

When adapting Stable Diffusion (Rombach et al., 2022) for multi-view generation, the previous approach (Shi et al., 2023a) changes the default scaled linear schedule into the linear schedule to emphasize more on early denoising stage for structural variation and view consistency. Inspired by this, we propose restricting the denoising time step of synthetic data during training. As small yet observable blur still exists in synthetic data with novel view generated by SV3D (Voleti et al., 2024), we limit them to training diffusion model only with large $t$. This restricts the backpropagation of these synthetic data to focus on the low-frequency component of the image like the overall structure and shape that faithfully follow text prompts and consistency between different views. Small $t$ values are only sampled on clear and physically consistent multi-view images rendered from Objaverse (Deitke et al., 2023) and supplemented high-quality 2D images from SA-1B (Kirillov et al.,

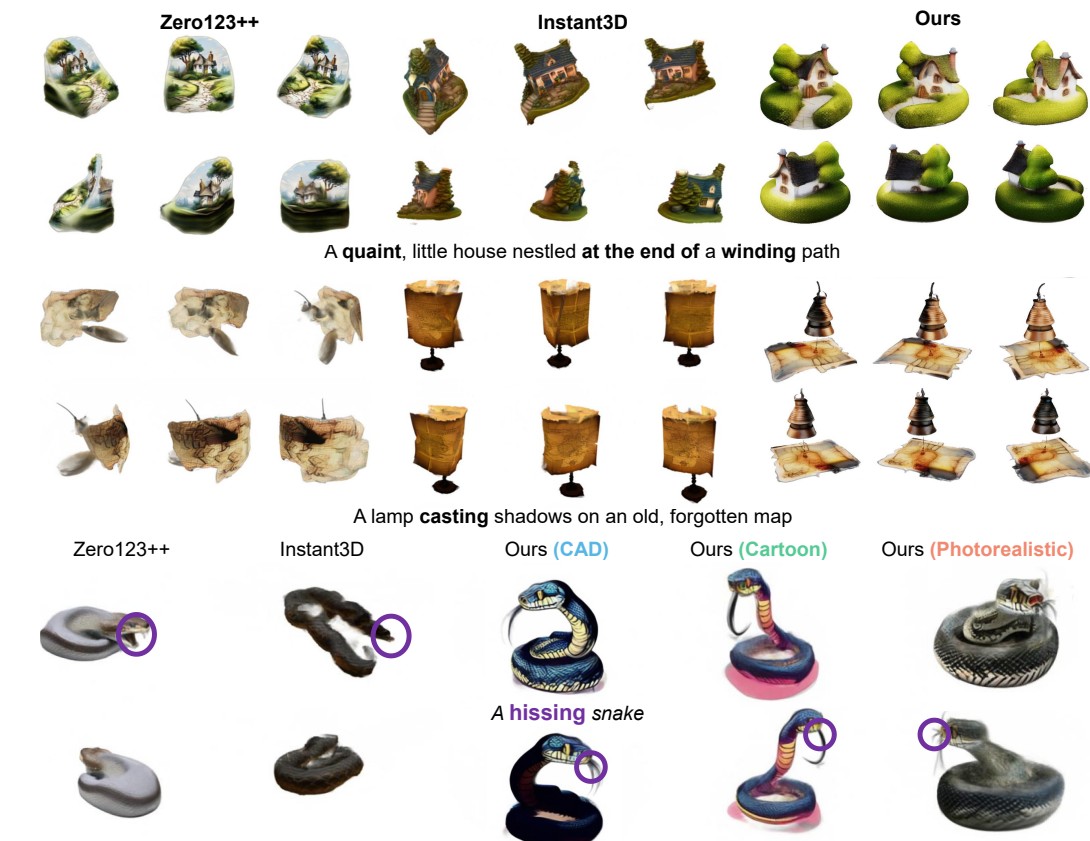

Figure 5: **Bootstrap3D generates 3D objects compared to other edge-cutting methods** given text prompt. More results with higher resolution are available in Sup.A.8.1.

2023), help model outcome high-quality images with more photo-realistic and varied texture details.

## 4 EXPERIMENTS

### 4.1 EXPERIMENT SETTINGS

**Training data.** For each set of 4-view images obtained from both Objaverse (Deitke et al., 2023) and generated by SV3D (Voleti et al., 2024) or Zero123++(Shi et al., 2023a), we use MV-LLaVA to generate long descriptive captions with predicted quality score. Detailed quality check of MV-LLaVA is supplied in Sup. A.4 and data analysis in Sup. A.3. In the end, we generate 200K 4-view image-text pairs on Objaverse (Deitke et al., 2023), 1000K 4-view image-text pairs from synthetic data from SV3D (Voleti et al., 2024) and Zero123++(Shi et al., 2023a). We also sample 35K HQ SA (Kirillov et al., 2023) data with captions from ShareGPT4V (Chen et al., 2023b).

**Training details.** We test our framework directly on the text-to-multi-view diffusion model. We fine-tune PixArt-$\alpha$ (Chen et al., 2023a) with backbone DiT-XL/2 (Peebles & Xie, 2023) model on the data as mentioned earlier. Similar to Instant3D (Li et al., 2023a), we train the diffusion model directly on 4-view images naturally arranged in a 2×2 grid. For 4 same view images from SA (Kirillov et al., 2023), we limit training time step $t \in [0, 50]$. We limit synthetic multi-view images $t \in [200, 1000]$. Regarding 3D object-rendered images, we do not limit $t$ but sample more frequently in the range $[50, 200]$ as a complement. We set the total batch size to 1024 with the learning rate set to 8e-5 for 20K steps. Training is conducted on 32 NVIDIA A100-80G GPUs for 20 hours with Flan-T5-XXL (Chung et al., 2024) text features and VAE (Kingma & Welling, 2013) features pre-extracted.

**Evaluation metrics.** We primarily benchmark the quantitative results of our approach and other methods from two main dimensions: 1). **Image-text alignment** measured by CLIP score and CLIP-

Table 1: **Benchmark of CLIP and FID score of text-to-multi-view (T2MV) models** on generated 4 view images, CLIP score tests on 110 text prompts from GPTeval3D (Wu et al., 2024) while FID is measured with the distribution of 30K object-centric images generated by SOTA T2I models. For text-to-image-to-multi-view(T2I2MV), we input I2MV models with single view images generated by Pixart-$\alpha$, which superior single view image quality is marked in green.

| Domain | Method | CLIP-R Score ↑ | | CLIP Score ↑ | | FID ↓ | |
|---|---|---|---|---|---|---|---|
| | | CLIP-L/14 | CLIP-bigG | CLIP-L/14 | CLIP-bigG | PG2.5 | PixArt-$\alpha$ |
| T2I | PixArt-$\alpha$ | 96.1 | 94.7 | 25.9 | 41.5 | 20.7 | 5.4 |
| T2I2MV | SV3D | 78.8 | 81.3 | 24.7 | 37.3 | 55.7 | 54.2 |
| | CRM | 77.5 | 85.1 | 24.9 | 38.9 | 59.0 | 52.2 |
| | Zero123++ | 78.0 | 84.5 | 24.2 | 36.9 | 53.2 | 49.3 |
| T2MV | Instant3D (unofficial) | 83.6 | 91.1 | 25.6 | 39.2 | 83.2 | 77.9 |
| | MVDream | 84.8 | 89.3 | 25.5 | 38.4 | 60.2 | 59.2 |
| | Bootstrap3D | **88.8** | **92.5** | **25.8** | **40.1** | **42.4** | **31.0** |

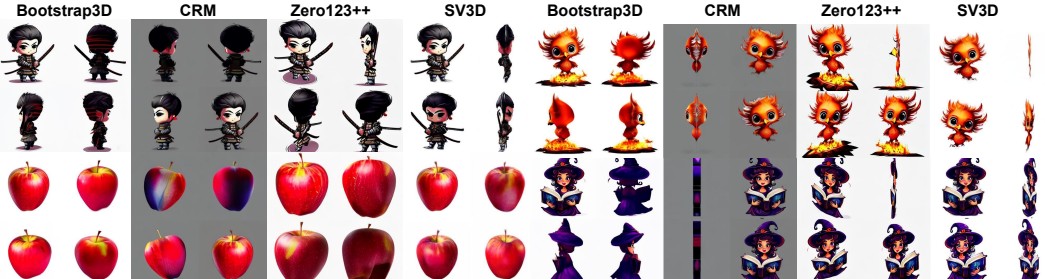

Figure 6: **Bootstrap3D can generate high quality multi-view images** in out of domain cases compare to other edge-cutting multi-view diffusion models trained on Objaverse only.

R score indicating the prompt follow ability of text-to-multi-view (T2MV) diffusion model. 2). **Quality of generated images** measured by FID (Heusel et al., 2017). Given the trend of decoupling multi-view image generation and sparse view reconstruction, we conduct tests separately on multi-view images by T2MV and rerendered images from generated 3D objects. To test the robustness and diversity of Bootstrap3D beyond prompts generated by GPT, we also collect real user prompts from public website, the details and test results are available in Sup. A.1.

**Evaluation details.** For CLIP-R Score and CLIP Score, we test on 110 text prompts from GPTeval3D (Wu et al., 2024) using different CLIP models (i.e., CLIP-L/14 (Radford et al., 2021) and CLIP-bigG (Ilharco et al., 2021)) following the same setting of Instant3D (Li et al., 2023a). Regarding the FID (Heusel et al., 2017) test, as there is no golden standard for HQ 3D objects, we follow the similar evaluation idea of PlayGround2.5 (Li et al., 2024) (PG2.5) to use powerful T2I model generated images to form ground truth (GT) distribution. We use curated prompts to guide powerful PixArt and PG2.5 to generate high-quality CAD-style images with a single object in the pure background. Rembg (etc, 2020) is adopted to create white background object-centric images. We use the method proposed in GPTeval3D (Wu et al., 2024) to generate 3K prompts. For both PG-2.5 and PixArt, we generate 10 images for each prompt with different seeds, resulting in 30K images to form the GT distribution of high-quality CAD-style objects.

**Comparing methods.** In addition to Instant3D (Li et al., 2023a) and MVDream (Shi et al., 2023b) as direct text-to-multi-view (T2MV) methods, we also adopt edge-cutting single image to multi-view (I2MV) methods CRM (Wang et al., 2024b), SV3D (Voleti et al., 2024) and Zero123++(Shi et al., 2023a). For these methods, we condition the diffusion model on the single view image generated by PixArt (prompted to generate CAD-style single object-centric image). The result of the CLIP score is 3 times averaged with different seeds. For FID, we use 3 different seeds for each of the 3K prompts to generate 9K images to test the distance with GT high-quality images.

## 4.2 EVALUATION OF MULTI-VIEW IMAGES

As illustrated in Tab.1, compared to other methods, the T2MV diffusion model trained by our framework yields the best results both according to image-text alignment and image quality. For qualitative experiments, we visualize some of the comparisons with other edge-cutting multi-view diffusion

model in Fig.6. For these image-to-multi-view models, we condition them on the top-left image generated by Bootstrap3D. Compared to these models trained solely on Objaverse Deitke et al. (2023), our model demonstrates superior generalizability when the image domain is beyond the domain of Objaverse. Since it is difficult to directly measure view consistency as there is no ground truth 3D object for text-to-3D generation. we evaluate the view consistency by synthesizing 3D objects through large reconstruction model in the following experiments. Qualitative results of real user cases are in Sup. A.1.

## 4.3 EVALUATION OF GENERATED 3D OBJECTS

Table 2: **Benchmark of CLIP and FID score of generated 3D objects** based on rendered 9 view images. *MVDream is tested on 200 generated objects for FID test using SDS (Shi et al., 2023b), other methods are tested on 1000 objects using GRM (Xu et al., 2024b) and InstantMesh (Xu et al., 2024a) as sparse view reconstruction model.

| Reconstruction | Method | CLIP-R Score ↑ | | CLIP Score ↑ | | FID ↓ | |
|---|---|---|---|---|---|---|---|
| | | CLIP-L/14 | CLIP-bigG | CLIP-L/14 | CLIP-bigG | PG2.5 | PixArt |
| SDS | MVDream* | 85.2 | 90.8 | **26.1** | 39.4 | 57.8 | 56.7 |
| GRM | Instant3D (unofficial) | 81.7 | 89.4 | 24.8 | 37.1 | 85.4 | 80.3 |
| | SV3D | 74.1 | 82.8 | 23.4 | 34.1 | 68.4 | 69.1 |
| | Zero123++ | 71.2 | 80.3 | 22.3 | 34.5 | 69.3 | 72.4 |
| | Bootstrap3D | 86.3 | 91.6 | 25.9 | **39.7** | **51.2** | **50.7** |
| InstantMesh | Zero123++ | 73.2 | 84.1 | 23.0 | 37.2 | 82.3 | 88.8 |
| | Bootstrap3D | **87.1** | **92.0** | 26.0 | 39.2 | 61.2 | 55.3 |

View consistency is another crucial factor in reconstructing reasonable 3D objects. Miss alignment between different views can lead to blurred areas in reconstructed objects by large reconstruction model (Hong et al., 2023; Wei et al., 2024). This misalignment causes a significant deterioration in quality, resulting in a notable increase in metrics like FID. To assess the view consistency directly on 3D object, we employ GRM (Xu et al., 2024b) and InstantMesh (Xu et al., 2024a) to reconstruct the object given sparse view images generated in Sec. 4.2. We render 9 view images evenly in orbit for each object and evaluate the image-text alignment and image quality. As reported in Tab. 2. Bootstrap3D, after conditioning GRM or InstantMesh on 4 view images, can generate the best 3D objects both according to image-text alignment and image quality. GPT-4V based human-aligned evaluation based on GPTeval3D (Wu et al., 2024) is supplied in Sup. A.6.

We also present visualizations of some results in Fig.5. Bootstrap3D can generate objects with higher quality and prompt following ability. For other methods, as shown in the first column of Fig.5, although the first image may be well aligned with the given text prompt, the final 3D object may be compromised due to the limitations of its poor generalizability as they are also fine-tuned on Objaverse (Deitke et al., 2023) only. More visualizations and discussions of this are in Sup. A.2

## 4.4 ABLATION STUDY

**Training Timestep Reschedule (TTR)** is proposed in 3.3 to better integrate different types of data. The training time step of synthetic data is restricted in $[T, 1000]$, where $T$ is a hyper-parameter to be set in training. We demonstrate the effect of the time-step limit in Fig.7, where the bar in the middle is the value of $T$. When $T$ is large, namely synthetic data won't affect more time-step at the end of the denoising process, Synthetic data has less influence on the denoising process towards the end, which leads to better view consistency but lower prompt-following ability. Conversely, if $T$ is small, the denoised result better follows the given text prompt but blurring becomes much more severe. In summary, there is a trade-off in injecting synthetic data into the training process: better image-text alignment comes at the cost of worse view consistency and increased blurring. Ultimately, we set $T = 200$ based on empirical study.

**Synthetic data and dense captioning** are proposed in our work to achieve high-quality images and better image-text alignment. We ablate their effects and the importance of data quantity in Tab. 3. Direct use of synthetic data without Training Timestep Reschedule (TTR) can cause severe blurs and deformation in final outcome. With the help of TTR, the mixture of data can not only improve image-text alignment but also maintain view consistency. Replacing Cap3D (Luo et al., 2024)'s caption

Table 3: **Ablation study of proposed components and quantity of synthetic data.** with CLIP-R Score represents image-text alignment and FID represents image quality.

| Methods | Multi-view Image | | Generated Object | |
|---|---|---|---|---|
| | CLIP-R Score | FID PG-2.5 | CLIP-R Score | FID PG-2.5 |
| Instant3D (unofficial) | 83.6 | 83.2 | 81.7 | 85.4 |
| Cap3D only | 77.9 | 101.3 | 74.6 | 120.4 |
| Cap3D + Synthetic Image (100k) w/o TTR | 81.5 | 92.0 | 71.2 | 134.6 |
| Cap3D + Synthetic Image (100k) w/ TTR | 83.3 | 60.8 | 80.2 | 70.6 |
| Dense recaption + Synthetic Image (100k) | 87.4 | 50.2 | 85.1 | 50.9 |
| Dense recaption + Synthetic Image (500k) | 88.8 | 42.4 | 86.3 | 51.2 |

A cat with two different colored eyes

A collection of fresh vegetables arranged in a wicker basket

Figure 7: **Ablation study of training time reschedule (TTR)** demonstrates a trade-off between image-text alignment and image quality with different $t$.

with MV-LLaVA's dense descriptive caption further improves the model's capability of following prompts faithfully. Improvement through increasing volume of data also proves the scalability of our framework.

## 5 CONCLUSION AND DISCUSSION

In this work, we introduce a novel framework that employs MLLMs and diffusion models to synthesize high-quality data for bootstrapping multi-view diffusion models. With a powerful fine-tuned 3D-aware MLLM serving as the dense captioner and quality filter, the generated synthetic data addresses the issue of insufficient high-quality 3D data. The proposed strategy of injecting different data at different training time steps uses the property of the denoising process to further achieve higher image quality while maintaining view consistency. We believe this work will contribute to the goal of achieving 3D content creation with each rendered view comparable with the single view diffusion model, with more advanced MLLMs and diffusion models on the horizon.

**Limitations and future work.** Despite its promise, our work still faces several unresolved challenges. Firstly, the multi-view diffusion model is only the first step of the 3D content creation pipeline. Sparse view reconstruction models also need improvement as most edge-cutting sparse view reconstruction models are also trained on Objaverse Deitke et al. (2023) only. Secondly, Although MLLMs can estimate general quality and view consistency, subtle view inconsistency is hard to detect until ambiguity leads to blurred areas in reconstructed 3D object. While the proposed Training Timestep Reschedule can mitigate this problem, it cannot solve the problem fundamentally. Using synthetic data to train sparse view reconstruction models and quality estimation directly based on the reconstructed object are thus interesting future directions for improving 3D content creation.

## 6 ETHICS STATEMENT

Our training code is modified based on public available repository `https://github.com/PixArt-alpha/PixArt-alpha`. Part of training data are synthesized by our proposed data generation pipeline. For other part of original Objaverse Deitke et al. (2023) data, we only use Cap3D Luo et al. (2024) filtered assets (Objects with CC BY-NC-SA and CC BY-NC licenses are removed, while we retain those with CC-BY 4.0, CC BY-SA, and CC0 licenses) and with face recognizable objects filtered through MSFW classifier and face detector. The ethical filtering in Cap3D make our work using only data without ethics problem. For our synthetic new data, We will launch both the generated captions for Objaverse Deitke et al. (2023) and high-quality synthetic data, model checkpoints and codes with CC-BY 4.0 license for the research community.

## 7 REPRODUCIBILITY STATEMENT

Main experimental setting/details (training data, hyperparameters, optimizer, evaluation settings, etc) are clearly presents in Sec. 4.1. For main results, we detail the full test settings in Sec. 4.1. For GPT-4V OpenAI (2023a) based preference study, we provide detailed test prompts and test settings in Sup. A.6. Readers can easily follow the same settings and reproduce all of our experiment results. We provide code for generating synthetic data. Both codes for training the model and testing are also available in supplementary material. The full data and model checkpoints are too large to provide public link without violation of double-blinding. We will release full data and model checkpoints after review.

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

# A APPENDIX

## A.1 EVALUATION ON WILD PROMPTS FROM REAL USERS

The results of the main part of the paper are only tested on GPT generated prompts. To test our work's capability in wild cases, we also collect real user prompts and compare our method with Instant3D (Li et al., 2023a). specifically, we randomly collect 100 prompts from `https://www.meshy.ai/` and test the CLIP-R precision as well as GPT based evaluation (detailed in Sup. A.6). Results and some qualitative cases are shown in Tab. 4 and Fig. 8. We highlight that our Bootstrap3D excels Instant3D (Li et al., 2023a) when tested on real user prompts through training on synthetic data.

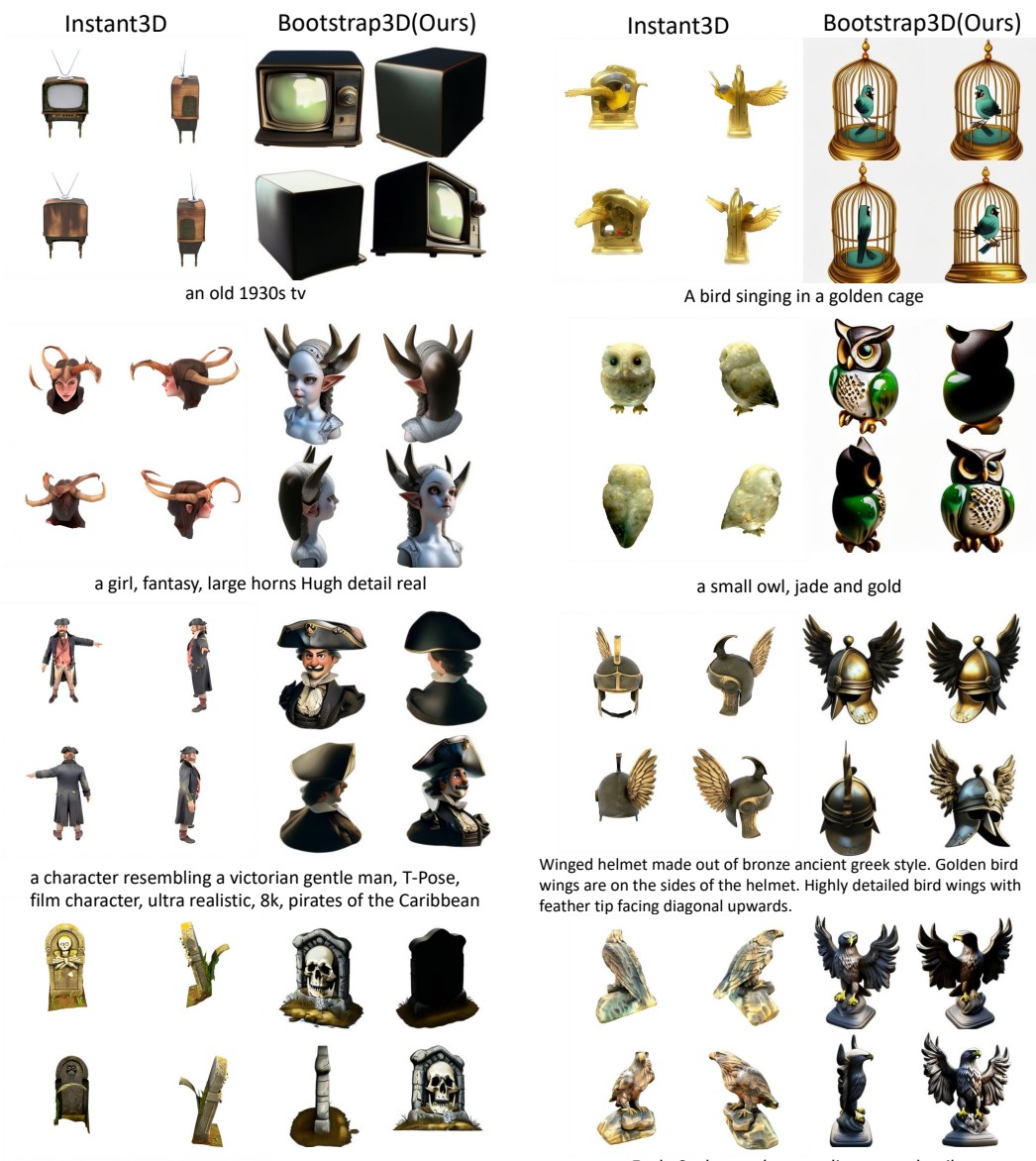

Figure 8: **Real user prompt cases** visualization compared to Instant3D (Li et al., 2023a)

## A.2 MORE VISUALIZATION COMPARED TO OTHER METHODS.

We show more visualization of the quantitative experiments shown in the main paper in Fig.9

Table 4: **Test results of in the wild cases.** Bootstrap3D also excels Instant3D (Li et al., 2023a) in generating high quality images according to real user prompts.

| Method | CLIP based metric CLIP-R score | GPTEval3D image-text alignment | texture detail |
|---|---|---|---|
| Instant3D (unofficial) | 77.0 | 22.0% | 24.5% |
| Bootstrap3D | 83.5 | 78.0% | 75.5% |

A compact, cylindrical, vintage pepper mill, with a polished, ornate **brass** body, slightly worn from use, placed **beside a porcelain plate** on a **checkered tablecloth**

A velvet-lined violin case, which opens to reveal a garden of miniature roses

A stone bridge arching over a babbling brook, encrusted with moss and echoing with stories, **cartoon style**

Figure 9: **Generated multiview images compare to other methods.** Our method can generate multi-view images with long text control without encountering blurring effect from data generated by SV3D thanks to TTR and quality filtering.

For Image-to-3D methods, they can sometimes produces significant motion blurring and fails when the input image is out-of-distribution (like the 3rd cartoon style case). We resample the high-quality segment of the distribution of generated images using quality filtering based on MLLM methods. Furthermore, by employing TTR, we limit the impact of these data when training multi-view diffusion models, allowing our model to produce much clear results. In addition, we use a caption rewriting method, enabling finer prompt control for the generated multi-view images.

## A.3 DATA STATISTICS

### A.3.1 CAPTION ANALYSIS

Fig. 10 and 11 provide a visualization of the root noun-verb pairs for the captions generated by GPT-4V (OpenAI, 2023a) and MV-LLaVA. It's clear to see that the diversity and linguistic expression of the captions produced by MV-LLaVA are highly matched with those of GPT-4V. We believe the highly detailed description focusing on object's texture, shape and color have potential usage beyond training multi-view diffusion model in the field like object texturing (Fang et al., 2024) and stylization (Sharma et al., 2023) in Computer Graphics. MV-LLaVA can also serve as free and efficient 3D object assistant comparable with GPT-4V for future research of 3D content creation.

Fig. 12 visualizes the histogram of caption length compared with Cap3D (Luo et al., 2024). We fine-tune MV-LLaVA to generate two different lengths suitable for different diffusion architecture, namely CLIP-based text encoding (Blattmann et al., 2023; Podell et al., 2023) with 77 token length and T5 based text encoding (Chen et al., 2023a; 2024a) with 120 token length. Both excel the length of Cap3D with less hallucinations.

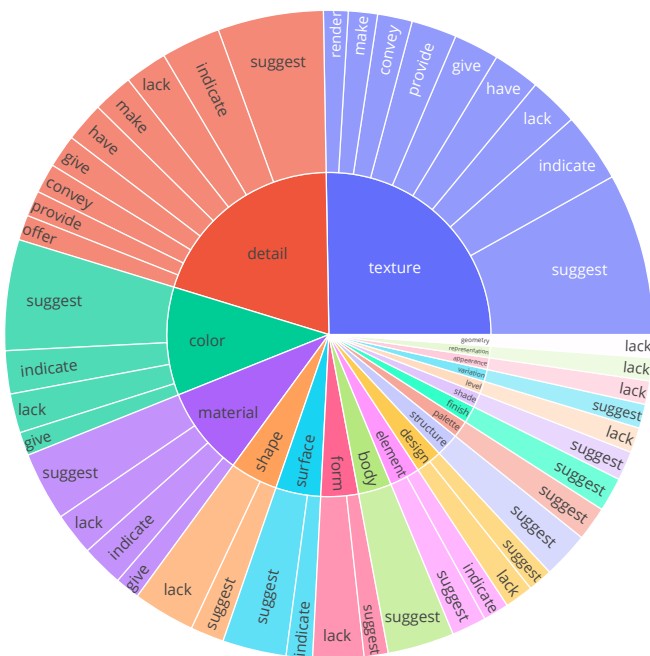

Figure 10: **Visualized analysis of dense reasoning descriptions generated by GPT4-Vision (OpenAI, 2023a)** of the root noun-verb pairs (occurring over 1%) of the descriptions

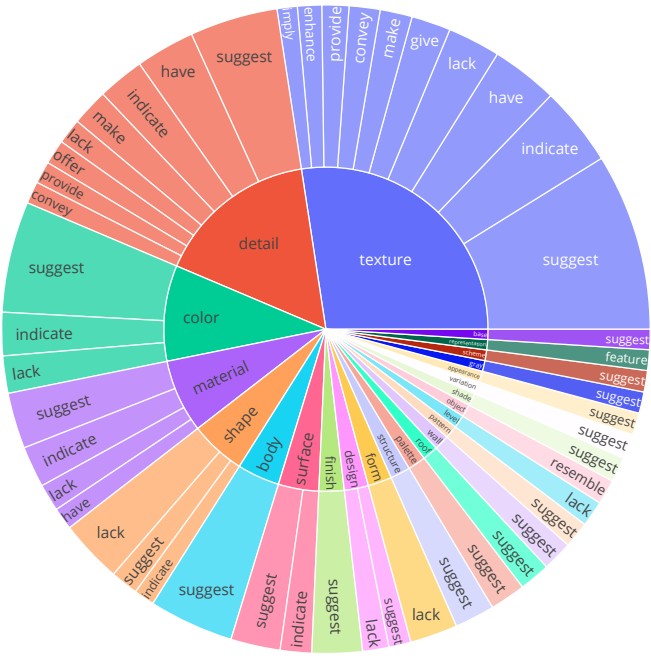

Figure 11: **Visualized analysis of dense reasoning descriptions generated by our MV-LLaVA** of the root noun-verb pairs (occurring over 1%) of the descriptions

### A.3.2 ESTIMATED QUALITY ANALYSIS

For direct grasp of the quality of objaverse data and synthetic data used to train diffusion model, we randomly picked some of multi-view images from different score rank. Results are shown in

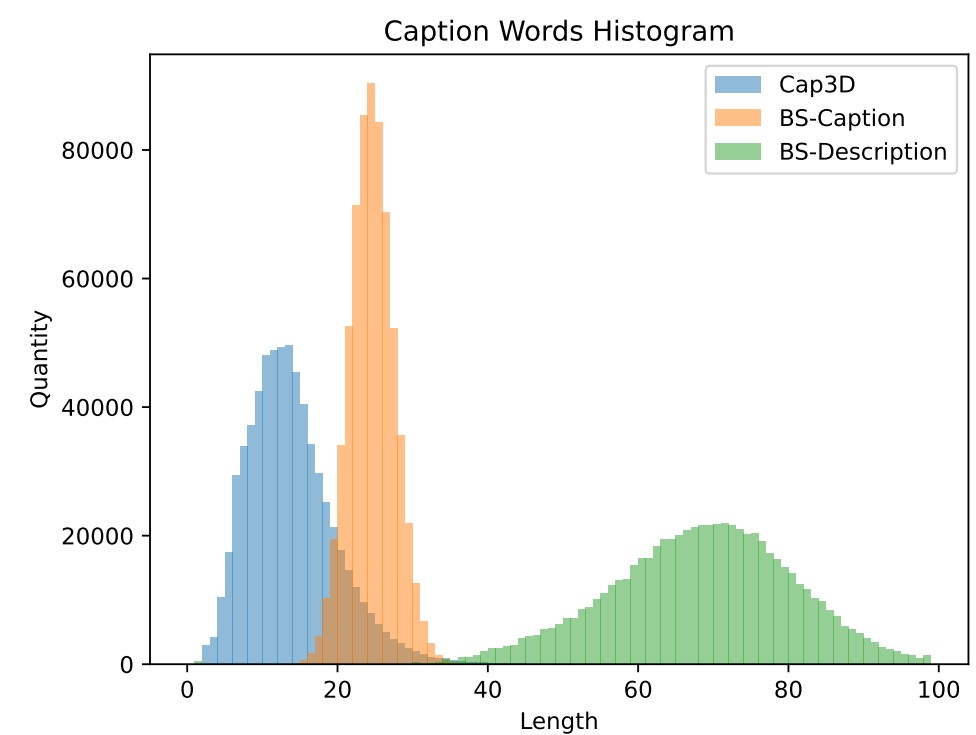

Figure 12: **Histogram Visualization of the Caption Length** compared with Cap3D (Luo et al., 2024)

Table 5: **Comparison of lexical composition of the captions** generated by GPT4-Vision and Share-Captioner.

| Lexical | n. | adj. | adv. | v. | num. | prep. |
|---|---|---|---|---|---|---|
| GPT-4V (OpenAI, 2023a) | 29.1% | 16.0% | 1.5% | 11.1% | 0.5% | 9.0% |
| BS-Description | 28.5% | 16.0% | 1.4% | 10.8% | 0.3% | 8.6% |
| BS-Caption | 30.2% | 23.0% | 0.3% | 5.6% | 0.1% | 8.9% |

Fig. 16, Fig. 17 and Fig. 18. We use high quality data with score 4 and 5 for the training of multi-view diffusion model.

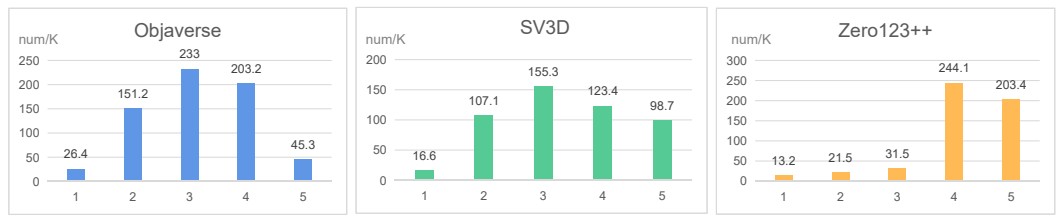

Figure 13: **Quality score statistics of different data source.**

We count the number of multi-view images from different data sources, namely 660K from Objaverse, 500K from SV3D (Voleti et al., 2024) and 500K from Zero123++ (Shi et al., 2023a) generated by our Bootstrap3D pipeline. Result are shown in Fig.13. For Objaverse and SV3D, the assigned score is normal and we use score 4 and score 5 multi-view images as high quality data for training.

However, for Zero123++, most objects are assigned with score greater than 3. We attribute this phenomenon to the fact that Zero123++ tend to generate objects with less motion blurring but more stretching and deformation compared to SV3D. Joint training of MV-LLaVA on three different data source lead to higher and more focused distribution for Zero123++'s multi-view images. For this part of synthetic data, we leave only score 5 multi-view images as high quality data.

## A.4 QUALITY OF MV-LLAVA

### A.4.1 CHOICE OF NUMBER OF UNFROZEN LAYERS OF VISION ENCODER.

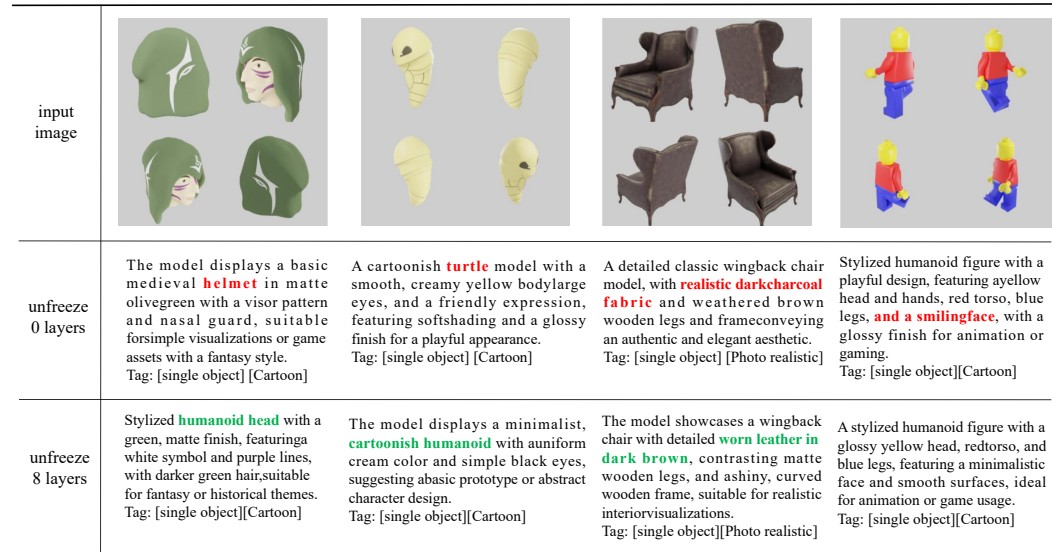

Figure 14: **Qualitative results of unfreeze final layers of CLIP (Radford et al., 2021) vision encoder** compared to original fixed vision encoder setting in LLaVA (Liu et al., 2024a).

Inspired by ShareGPT-4V (Chen et al., 2023b), we unfreeze selected final layers of the CLIP (Liu et al., 2024a) vision encoder during the initial phase of vision language alignment. The CLIP-L/14 model used for LLaVA (Liu et al., 2024a) contains 24 transformer layers. We selectively unfreeze some of final layers to enable the CLIP model to focus more on details such as texture of multi-view images. After qualitative manual screening, we select to unfreeze eight layers to yield better results. Fig. 14 illustrates the differences between unfreezing eight layers and not unfreezing any (the original training setting of LLaVA (Liu et al., 2024a)). The red sections highlight the erroneous hallucinations occurring when the vision encoder remains fully unchanged, while the green sections indicate accurate descriptions of the image content. This demonstrates that partially unfreezing the vision encoder can produce more precise captions and reduce some hallucinations.

### A.4.2 QUANTITATIVE QUALITY STUDY

To test the quality of our MV-LLaVA. We propose two quantitative study over the quality of captions and the alignment of quality estimation with human experts. In first study, we randomly picked 200 object from Objaverse (Deitke et al., 2023) and exclude training data of MV-LLaVA. We use GPT4-V (OpenAI, 2023a) and MV-LLaVA to generate descriptive captions for each object. We invite human volunteers to choose their preference over shuffled captions. Results are shown in Tab. 6, where MV-LLaVA shows comparable captioning ability with powerful GPT4-V (OpenAI, 2023a), which is essential to generate millions of high quality image-text pairs for the training of text to multi-view image diffusion model.

Second experiment studies MV-LLaVA's ability in quality estimation of both 3D assets and generated multi-view images. We invite human volunteers to estimate the quality of multi-view images rendered from Objaverse (Deitke et al., 2023) or generated by SV3D (Voleti et al., 2024). As there is no golden standard for multi quality classification, We ask them to separate the randomly select

multi-view images into approximately two half and serve as GT quality. We use MV-LLaVA to esti-mate the quality of these images and generate confusion matrix. Results are shown in Tab.7. Given the great amount of source data of 3D assets and infinite synthetic data, we care more about the false positive rate, as these data will be mixed into training data. In this observation, we highlight the false positive rate of over 20% for SV3D (Voleti et al., 2024) generated multi-view images. This result align with the observation of inevitable motion blurring of SV3D (Voleti et al., 2024). To leverage this part of data source for data diversity without hurting the final quality. We propose Training Noise Reschedule to avoid samplings from these synthetic data when time step is small.

Table 6: **Human evaluation** on the quality of generated captions from MV-LLaVA vs. GPT4-Vision (OpenAI, 2023a) over 200 validation samples from Objaverse (Deitke et al., 2023).

| Preference | GPT4-Vision (OpenAI, 2023a) | MV-LLaVA | Comparable |
|---|---|---|---|
| Percentage | 39.5% | 34.5% | 26.0% |

Table 7: **Confusion matrix** of mutli-view images quality estimation.

| Objaverse quality check | | | Synthetic quality check | | |
|---|---|---|---|---|---|
| | HQ-gt | LQ-gt | | HQ-gt | LQ-gt |
| HQ by model | 31.0% | 4.5% | HQ by model | 34.5% | 11.5% |
| LQ by model | 11.0% | 53.5% | LQ by model | 17.0% | 37.0% |

### A.4.3 QUALITATIVE CAPTION QUALITY STUDY

We selective compare some of the captions generated by Cap3D (Luo et al., 2024) and MV-LLaVA in Fig. 15. Our MV-LLaVA can generate more detailed descriptive captions with less hallucinations.

**Score: 1**

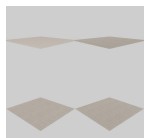

**Ours:** The model shows a plain with nearly identical flat surfaces with a uniform beige color, lacking any distinctive features or textures. There are no discernible details indicating what the surfaces represent or their potential use, and thus the model serves as an extremely basic representation of a flat object.

**Cap3D:** 3D rendering of a square, tiled floor and ceiling.

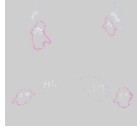

**Ours:** The model is a highly abstract representation, possibly of a figure or object with an unclear purpose. It features thin, rod-like lines and sparse clusters of pink and blue colors, lacking any discernible texture, material quality, or detailed features.

**Cap3D:** A collection of drawings featuring a jellyfish, a girl with various accessories, a fish, and a unicorn.

**Ours:** The model is composed of faint, scattered dots and lacks discernible shape or texture, making it difficult to determine its intended representation. It appears as a collection of sparse, flat details without any clear function or recognizability.

**Cap3D:** A 3D object featuring elements of maps, a fish, a heart, a person, and a ghost, all adorned with pink and blue dots.

**Score: 2**

**Ours:** The model represents a minimalistic art gallery scene with discernible but simplified elements. Semi-transparent white walls and floors suggest a clean, empty space. Slight color variations hint at lighting nuances. Sparse objects including picture frames are rendered in simplistic grayscale, lacking texture but offering a rudimentary representation of an art display.

**Cap3D:** A 3D rendering of a white exhibition space featuring a round table, pictures, a water feature, and a chair.

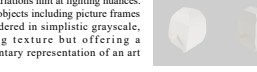

**Ours:** The model is a low-polygon representation of a character's head with a simplified form, featuring a broad, flat surface with minimal texture. The main body of the model is white, while the facial features are depicted in a flat grey tone, lacking depth and detail. It suggests a basic, monochromatic material, possibly indicative of clay or a digital sculpting tool...

**Cap3D:** A low poly 3D model of a white ghost, small bird, and teddy bear.

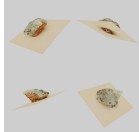

**Ours:** 'The model represents a rock-like object with a textured surface, depicting moss or lichen patches in shades of green and gray. The underlying stone surface exhibits rust tones suggesting iron-rich sedimentary rock. However, the model lacks finer details such as specific rock strata or additional variations that could enhance its realism.

**Cap3D:** 3D models of a fruit, paper airplane, rock, and green leaf with various surfaces and interactions.

**Score: 3**

**Ours:** The 3D model represents a minimalist architectural form, possibly a building or pavilion with an external staircase. It's composed of white and grey tones, where white suggests a smooth concrete or plaster and grey represents metallic or stone features. The model lacks intricate textures but has discernible geometric shapes and shadows, offering a basic sense of depth and spatial relationships.

**Cap3D:** A 3D rendering of a white cabinet, staircase, small box, chair, desk, ceiling-mounted light fixture, and ceiling panel.

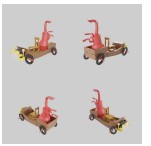

**Ours:** The model represents a whimsical, simplified vehicle consisting of a brown wagon with yellow wheels and a red, toy-like trailer being pulled. The wagon's body has a flat, wood-like texture, while the trailer's body shows a smooth, glossy finish. The handle and wheels of the trailer are colored in bright yellow, enhancing its toy-like appearance...

**Cap3D:** A 3D wooden toy car model featuring a stove, chair, pipe, spout, bottle, and steering wheel.

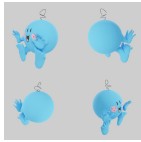

**Ours:** The model is a whimsical, cartoon-style character with a spherical blue body and exaggerated, bulbous limbs. The facial features include a wide, cheerful smile and large, expressive eyes with a simple white sclera and black pupils. The limbs' ends have been stylized into points, enhancing the playful nature of the model...

**Cap3D:** A 3D blue sphere with pink eyes and hands, resembling a cartoon character.

**Score: 4**

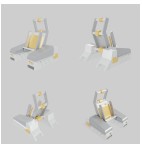

**Ours:** This model portrays a small, stylized scene of a Japanese garden with a pond and cherry blossom trees. The pond is depicted in a tranquil shade of blue, surrounded by vibrant green grass. Pink cherry blossom trees, a symbol of spring in Japan, are scattered around the pond with varying shades of pink. A small bridge with a red railing crosses the pond, adding a pop of color to the scene. The textures are simple yet distinct, with the grass and trees having a flat appearance...

**Cap3D:** 3D model of a Japanese garden village with trees, a bridge, and a hill, featuring a plane flying overhead.

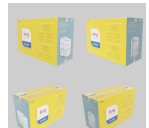

**Ours:** This model is a robotic chair with a minimalist design, rendered in a clean, modern style. The seat and backrest are white, implying a smooth, possibly plastic or painted metal surface. The mechanical arm and base are a metallic gray, suggesting a metallic texture with a matte finish. Gold-colored accents, likely meant to represent controls or adjustments, add a touch of color to the model...

**Cap3D:** A 3D model of a white, yellow, and gold machine-like spaceship.

**Ours:** The model is a packaged product, most likely an air purifier, with a recognizable brand and model identifiable from the images. The packaging is a vibrant yellow with white accents, which gives a sense of a plastic or laminated paper material. The product image on the box shows a white device with a blue air filter, conveying a clean and modern aesthetic...

**Cap3D:** Yvy toilet paper box, Yui air purifier, Yoyo toner cartridges, Yii vitamins, and Yui sanitary napkins in yellow and white packaging with respective logos.

**Score: 5**

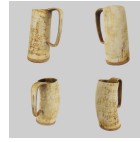

**Ours:** This 3D model represents a wooden tankard with a detailed wood grain texture in natural tan shades, featuring realistic knots and variations in the wood. The handle and rim of the tankard are well-defined, and the model exhibits a rustic, handcrafted appearance. The carved lines and wood texture give it an authentic look...

**Cap3D:** A 3D horn-shaped jug with a wooden handle and writing on it.

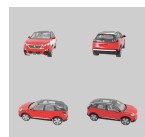

**Ours:** The model is a red SUV with a high level of detail, including realistic shapes for the body, windows, and wheels. The body has a glossy finish, simulating automotive paint, while the black trim and silver rims have a matte texture. The wheels are detailed with intricate silver spokes that resemble alloy wheels, and the tires are a vibrant green with tread patterns adding to the model's realism...

**Cap3D:** Red Peugeot 208 SUV 3D model.

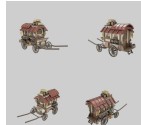

**Ours:** This 3D model is a charming, medieval-style vendor cart with a high degree of texture detail, suggesting materials like weathered wood for the main structure, rusty metal for the fittings and wheel rims, and tattered fabric for the awning. The cart is laden with goods such as sacks, barrels, and hanging items...

**Cap3D:** 3D model of a wagon with a roof and wheels, featuring a horse.

Figure 15: **Caption comparison with Cap3D (Luo et al., 2024).** Our MV-LLaVA can generate long captions that faithfully describing 3D assets from different perspectives like color, geometry and texture.

Score1

Score2

Score3

Score4&5

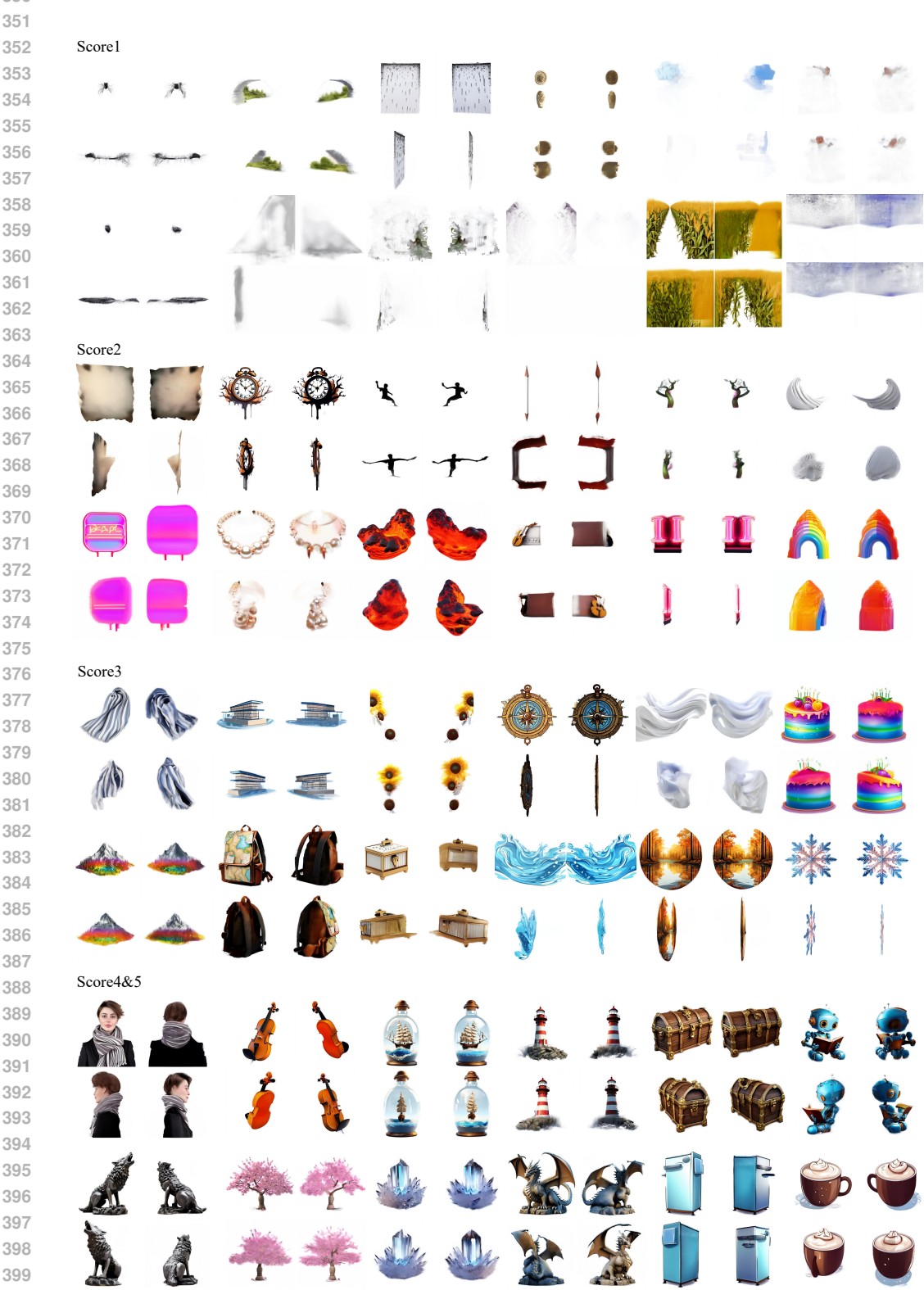

Figure 16: **Randomly picked multi-view images with different scores from 500k synthetic data generated by SV3D (Voleti et al., 2024).**

Score: 1

Score: 2

Score: 3

Score: 4&5

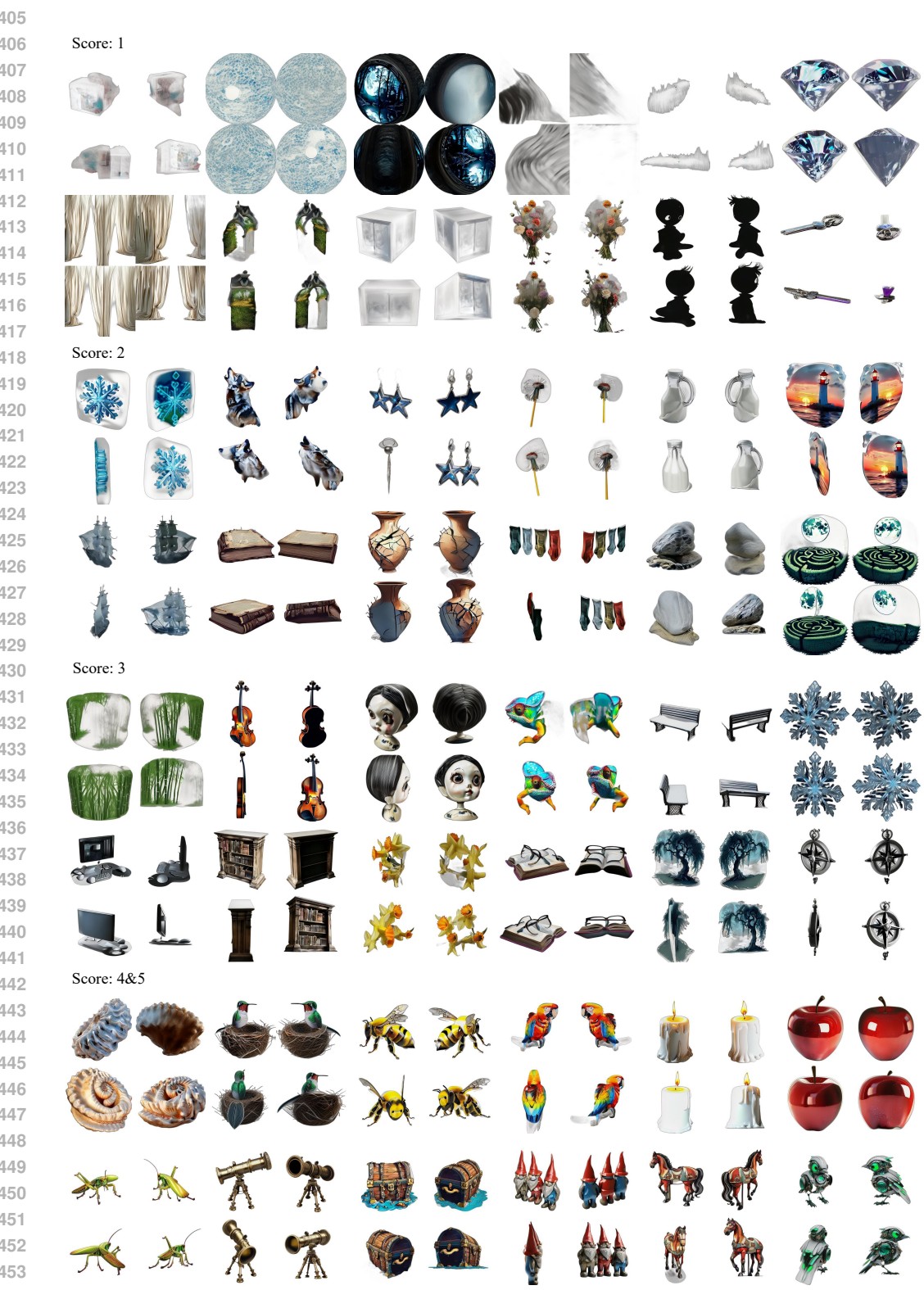

Figure 17: **Randomly picked multi-view images with different scores from 500k synthetic data generated by Zero123++ (Shi et al., 2023a).**

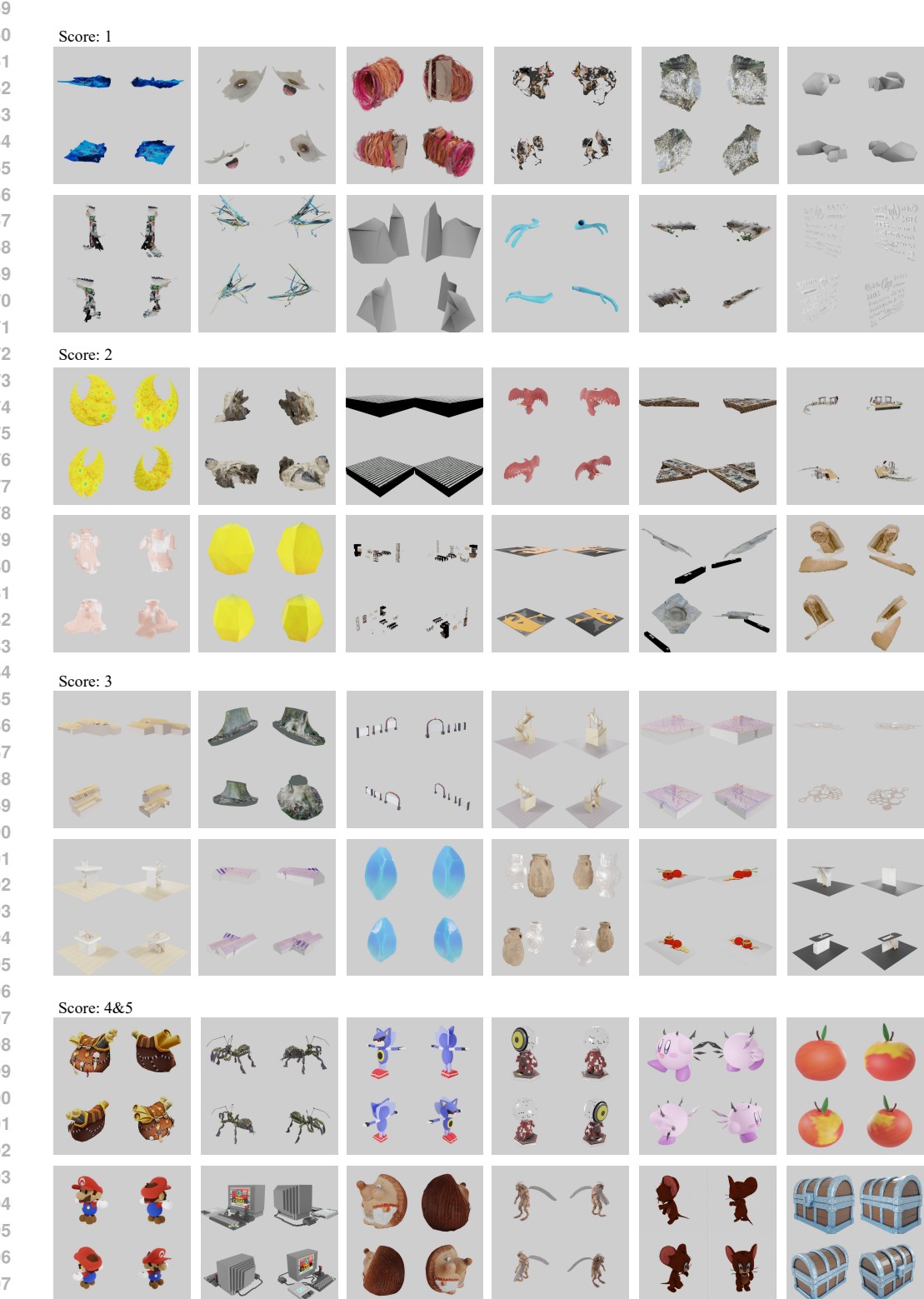

Figure 18: **Randomly picked multi-view images with different scores from 660k Objaverse (Deitke et al., 2023) 3D assets.**

## A.5 DETAILS OF PROMPT DESIGN

### A.5.1 PROMPTS FOR GPT-4V FOR QUALITY CHECK

Assume you are a quality checker of a diffusion model. This diffusion model is trained to achieve novel view synthesis. I give this model the image in the upper-left side and it generate novel views in the rest three images(upper-right, lower-left, lower-right). You should tell me the quality of the generated novel view images. The score ranges from 1 to 5, representing the quality of the model from low to high. The detailed evaluation criteria are as follows:

1. The novel views are difficult to discern what the image supposed to be, lacking in recognizability. It has no usable value.
2. The novel views are distinguishable, clearly determine what the object/scene is similar to the given ground truth image. However, there is obvious inconsistency between the novel view synthesized images and groud-truth image. There are many obvious areas of image is blurred or indicating rotation.
3. The novel views are relatively good, the inconsistency between novel view synthesized images with groud-truth image is not obvious. The blurring area indicating rotation or uncerntainty is acceptable for usage.
4. The novel views are pretty good, although the might be blurring areas or less resolution. the view consistancy is well maintained.
5. The novel views are excellent. It is hard to tell which image from four is ground-truth and which is synthesised.

You shoud give me the overall score with one score number, with reason in next line. besides the quality check, I need you to generate a long discriptive caption for the scene/object from 4 different view. focusing on the part/object relative position, color, number of objects and so on with no more than 50 words and no less than 30 words. DO NOT MENSHION MULTI-VIEW IMAGES FROM DIFFERENT PERSPECTIVE since it is a single scene/object. you should rearrange your result in a JSON format. if all the images(include the groud-truth image) are of low quality, just output a lowest score.

Here is an example for you:
{"score": 4, "reason": "The novel views generated from the model are quite convincing with a high degree of consistency in terms of texture, lighting, and color when compared to the ground-truth image. There is some minor distortion in shape and perspective, but the overall quality is high, and it maintains the realism of the scene.", "caption": "A cluster of shiny five apples, ranging from deep red to sunny yellow, sits comfortably within a rustic woven basket. Their smooth, round forms are grouped closely, reflecting light and casting soft shadows that accentuate their voluminous curves and vibrant colors."
}

This is a quad image generated from rendering a SINGLE 3D model FROM FOUR DIFFERENT views. I would like you to score the quality of this models to evaluate its current state. The score ranges from 1 to 5, representing the quality of the model from low to high. The detailed evaluation criteria are as follows:

1 point: The overall quality of the model is quite poor, making it difficult to discern what it is supposed to be, lacking in recognizability. The model is almost one solid block, or extremely scattered, or in fragments. It has no usable value.

2 points: The overall quality of the model is relatively poor, but it is possible to guess what it is, possessing low recognizability. It preliminarily has some geometric shape and can be considered a prototype model element, lacking identifiable material information, and almost has no usable value.

3 points: The overall quality of the model is average, it is possible to determine what it is, having certain recognizability. Different areas use different materials (colors), it preliminarily has usable value, and initially has aesthetic value.

4 points: The overall quality of the model is relatively high, it can be clearly determined what it is, with high recognizability. It preliminarily has certain texture details, and different parts of a model can be clearly distinguished, having high usable value and certain aesthetic value.

5 points: The overall quality of the model is extremely high, allowing for the classification of the model's type at a very fine granularity. It has high texture details, is a fully formed 3D model that can be used for games, simulations, or even animations, and has high aesthetic value.

After scoring, please also generate a description of the current model. If the model quality is low, only a brief description is needed; when the model quality is high, a complete description of the different details of the model is required. The description process should focus on color, material, texture details as much as possible. You can also recommended to suggest overall style. With NO MORE THAN 120 words. Especially discribe color and meterial of different parts concretely and faithfully, let the reader easilly imagine the same model.

Finally, I hope you can annotate two kinds of tags for the model. Tag1 is about the style of overall model. You can choose from [photo-realistic], [carton] and [CAD]. Tag model as [CAD] when seems like a preliminary work build by CAD software and not real. Tag model [carton] when it is good enough with carton style. Tage model [photo-realistic] when the model seems like real object in the world; Tag2 is about the scale that the model represents, you can choose from [single object], [multi-object], [small scene] and [large scene]. Assign model [large scene] when it represents scene like urban street, park, etc. Assign it as [small scene] when it represents scene like inner structure or design of a house, small area, etc. Assign it as [multi-object] when it represents combination of multi objects. Assign it as [single object] when it represents single object.

Here are three examples. You should follow this format:
e.g. 1
Score: 1
Description: The model depicts a very basic and abstract urban planning concept with indistinct structures and simplistic landscaping, lacking detail and texture, appropriate for early-stage design or conceptual visualization.
Tag: [Photorealistic] [large scene]
e.g. 2
Score: 2
Description: The object is a simple sphere with a homogeneous speckled texture, suggesting a stone-like material. The colors vary slightly between shades of dark gray, brown, and rust, with a matte finish. It lacks specific features or details that would indicate a higher level of complexity or function.
Tag: [Photorealistic] [single object]
e.g. 3
Score: 3
Description: The model appears to represent an architectural structure with two levels. Different colors suggest varied materials: translucent white for the structural framework, solid blue representing walls or glass panels, and yellow for interior elements, possibly stairs or floors. The style seems utilitarian, potentially for preliminary construction visualization.
Tag: [CAD] [small scene]
e.g. 4
Score: 4
The model depicts a metallic livestock handling equipment known as a cattle chute. It is rendered in shades of dark gray, conveying a metallic texture consistent with steel or iron. The structure is detailed with bolts, bars, and sliding gates, implying a sturdy construction. Text labels like \"METALCORP\" and \"CATTLE MASTER\" in blue enhance realism, suggesting a commercial quality model suitable for simulations or instructional material. The style is industrial and pragmatic.
Tag: [Photorealistic] [single object]
e.g. 5
Score: 5
The model is a stylized, anime-inspired character with a cheerful expression. Hair is rendered in a turquoise shade, contrasting with ribbons in alternate hues of pink and blue. Skin tone is in a soft peach, while the outfit combines white, grey, and gold tones, with a large yellow flower accessory. Surfaces show subtle shading, indicating variations in material. The playful, colorful appearance suggests a light-hearted, fantasy aesthetic.
Tag: [Cartoon] [single object]

Figure 19: **Prompt for GPT-4V to generate caption and estimate quality of multi-view images from SV3D (Voleti et al., 2024), zero123++ (Shi et al., 2023a) and Objaverse (Deitke et al., 2023).**

Detailed prompts are shown in Fig.19.

### A.5.2 PROMPTS FOR MV-LLAVA INSTRUCT TUNING

Table 8: **Instruct tuning prompt for SV3D (Voleti et al., 2024) and Zero123++ (Shi et al., 2023a) multi-view images**

| prompt type | prompt |
|---|---|
| generate caption | \<image\>\<image\>\<image\>\<image\>\nWhat is this multi-view photo about? generate a short caption for me. 
 \<image\>\<image\>\<image\>\<image\>\nGenerate a short caption of the following multi-view image. 
 \<image\>\<image\>\<image\>\<image\>\nCan you describe the main features of this multi-view image for me by a short caption? |
| reasoning | How about the view consistency of this synthesized multi-view image? 
 Do some comments about the view consistency of this synthesized multi-view image. 
 What do you think about the view consistency of this synthesized multi-view image? |
| quality estimation | What do you think about the overall quality of view consistency of three synthesized novel views? Choosing from "poor", "relatively poor", "boardline", "relatively good", "good", "perfect". |

Table 9: **Instruct tuning prompt for Objaverse (Deitke et al., 2023) rendered multi-view images**

| prompt type | prompt |
|---|---|
| long description | \<image\>\<image\>\<image\>\<image\>\nWhat is this multi-view photo about? generate a long descriptive caption for me. 
 \<image\>\<image\>\<image\>\<image\>\nGenerate a long descriptive caption of the following multi-view image. 
 \<image\>\<image\>\<image\>\<image\>\nCan you describe the main features of this multi-view image for me by a long descriptive caption caption? |
| caption | \<image\>\<image\>\<image\>\<image\>\nWhat is this multi-view photo about? generate a short caption for me. 
 \<image\>\<image\>\<image\>\<image\>\nGenerate a short caption of the following multi-view image. 
 \<image\>\<image\>\<image\>\<image\>\nCan you describe the main features of this multi-view image for me by a short caption? |
| quality estimation | What do you think about the overall quality of this 3D model? Choosing from "poor", "relatively poor", "boardline", "relatively good", "good", "perfect". |
| scale tag | What do you think about the scale of the 3D model represents? Choosing from "single_object", "multi-object", "small_scene", "large_scene". |
| style tag | What do you think about the overall style of the 3D model? Choosing from "CAD", "Cartoon", "Photo_realistic". |

### A.6 GPT-4V BASED 3D OBJECT GENERATION EVALUATION.

We adopt method proposed in GPTeval3D (Wu et al., 2024) for more thorough and human-aligned evaluation of the quality of generated object by different methods. A full test case is shown in Fig. 20. Left 9-view image is rendered from object generated by Bootstrap3D and the right one generated by Instant3D (Li et al., 2023a). We ask GPT-4V to mainly evaluate through comparison based on three dimensions: text-image alignment, low-level texture quality and 3D plausibility. The answer of GPT-4V shows its in depth perception ability of given reasonable comparison well

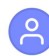

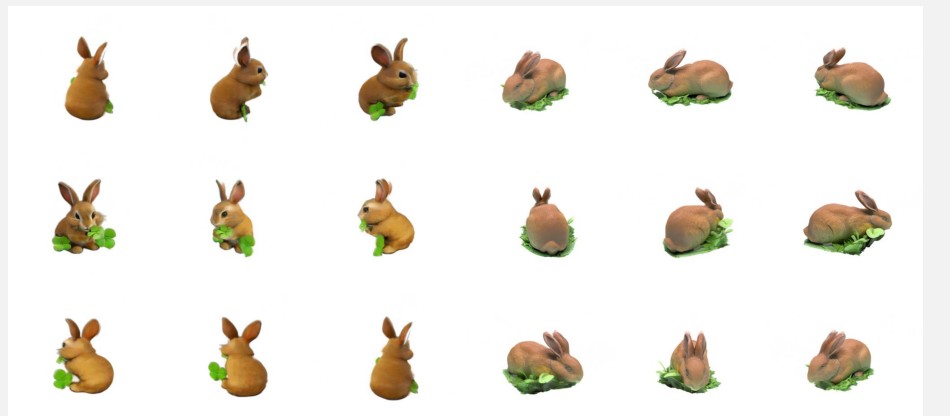

Our task here is the compare two 3D objects, both generated from the same text description. I will provide you with 9 specific view of two models, where the left part of it are image rendering and normal rendering of 3D object 1, and the right part denotes those of 3D object 2.

We want to decide which one is better according to the provided 3 criteria:

1. **Text prompt and 3D shape Alignment**. This evaluates how well the 3D model corresponds to the text description. Please first describe each of the two models, and then evaluate how well it covers ALL the attributes in the original text prompt.

2. **Low-Level Texture Quality**. Focus on LOCAL parts of the RGB images: whose texture is sharper, more realistic, with high resolution, and with more details? Remember that the local texture quality could be high even if the text-3d alignment is weak, this is independently evaluated.

3. **3D Plausibility**. Look at both the RGB and normal images and imagin a 3D model from the multi-view images. Which model is more normal, solid, and plausible? Decide if there are strange body problems, duplicated parts, any noisy or meaningless 3D structures.

Take a really close look at each of the images for each of the two 3D objects before providing your answer. When evaluating these aspects, focus on one of them at a time and do not let them affect each other. Make independent decisions between these criteria.

Provide a short analysis for each of the abovementioned evaluation criteria before giving your answer. The analysis should be very concise and accurate.

For each of the criteria, you need to make a decision using these three options:

1. Left (object 1) is better;
2. Right (object 2) is better;
3. Cannot decide.

IMPORTANT: PLEASE USE THE THIRD OPTION SPARSELY.

In the last row, summarize your final decision by "<option 1> <option 2> <option 3>".

Current text prompt is: **Brown rabbit nibbling on a clover leaf**

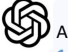

Analysis:

1. **Text prompt & 3D Shape Alignment**: The left model seems to align more closely with the text description, with a more defined **nibbling** action on the clover leaf and a more distinct brown color. The right model also shows nibbling on a leaf, but the action is not as pronounced, and the color of the rabbit appears **less brown** and **more uniform**. The left one is better.

2. **Low-Level Texture Quality**: The left model features sharper and more detailed textures **with visible fur strands and leaves details**. The right model has smoother and less detailed textures, especially noticeable on the rabbit's fur which appears less realistic. The left one is better.

3. **3D Plausibility**: Both models seem plausible and solid from a 3D perspective. However, the left one has **more natural poses and anatomically correct rabbit shapes**, with no awkward angles or duplicated parts. The right one has a less natural resting pose and slightly unusual proportions on the head and limbs. The left one is better.

Final answer:

1 1 1

Figure 20: **A test conversation with GPT-4V (OpenAI, 2023a)** of evaluating generated objects.

aligned with human preference. We thus choose to use GPT-4V rather than human volunteers to give reasonable evaluation.

We adopt the 110 test prompts proposed in GPTeval3D (Wu et al., 2024) to test Bootstrap3D generated object comparing with Instant3D (Li et al., 2023a), Zero123++ (Shi et al., 2023a) and MV-Dream (Shi et al., 2023b). For each methods, we conditioned model based on 110 test prompts with

4 different seeds, with each methods generates 440 objects, we make 1-to-1 comparison following aforementioned test setting. Results are reported in Tab. 10. Except MVDream (Shi et al., 2023b) (SDS) (which generates single object consuming 30 mins while Bootstrap3D only need 5 seconds.). Bootstrap3D excels in all three evaluation dimensions, which proves the ability of Bootstrap3D in creating high quality 3D objects.

Table 10: **GPT-4V based evaluation result.** the result is in format of "number of objects preferred geneated by Bootstrap3D/ that of other methods". Cases when GPT cannot answer the question or generates "cannot decide" answer are excluded.

| | Image-text alignment | Texture quality | 3D plausibility |
|---|---|---|---|
| Compared to Instant3D (Li et al., 2023a) (unofficial) | 247 / 116 | 202 / 162 | 259 / 110 |
| Compared to Zero123++ (Shi et al., 2023a) | 192 / 143 | 210 / 161 | 231 / 139 |
| Compared to MVDream (Shi et al., 2023b) (GRM) | 290 / 71 | 245 / 131 | 284 / 102 |
| Compared to MVDream (Shi et al., 2023b) (SDS) | 188 / 155 | 173 / 190 | 192 / 150 |

## A.7 IMPROVING DIRECT 3D GENERATIVE MODELS

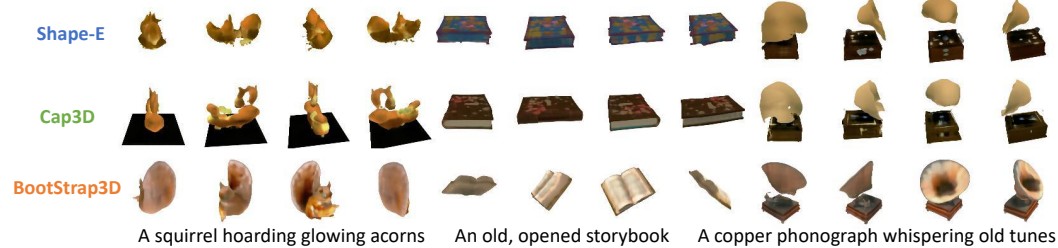

A squirrel hoarding glowing acorns    An old, opened storybook    A copper phonograph whispering old tunes

Figure 21: **Fine tuned Shape-E generation results** that shows better object-text alignment than original Shape-E (Jun & Nichol, 2023) and finetuned version in Cap3D (Luo et al., 2024).

Table 11: **Test results on Shape-E.** More accurate and descriptive 3D caption help model to achieve better object-text alignment.

| Method | FID ↓ | CLIP score ↑ | CLIP-R-precision ↑ |
|---|---|---|---|
| Shape-E | 37.2 | 80.4 | 20.3 |
| Cap3D | 35.5 | 79.1 | 20.0 |
| Ours | 35.3 | 81.2 | 22.1 |

In addition to fine-tuning the multiview diffusion model, we also evaluate our framework on direct 3D generative models, circumventing the use of multi-view images as intermediaries. For this purpose, we selected the Shape-E (Jun & Nichol, 2023) model for experiment and assess the outcomes following the testing method the same to Cap3D (Luo et al., 2024). Specifically, we fine-tune Shape-E using 250K BS-Objaverse data, ensuring that all entries scored greater than 3, accompanied by more precise and descriptive captions. The metrics for training and testing are consistent with those employed in Cap3D (Luo et al., 2024). Some qualitative results are presented in Fig.21, where our finetuned verson can generate object that follow text prompt more precisely. Quantitative results are detailed in Tab.11, where more accurate and desciptive captions than Cap3D can significantly improve metrics like CLIP score. Our findings indicate that improved data quality can significantly enhance object-text alignment and visual quality of Shape-E. This experiment substantiates that our pipeline, characterized by detailed captions and quality filtering, is also effective for direct 3D objects generation represented by neural field.

**A chibi phoenix reborn from ashes, flames gently flickering around it**   **A cracked teapot heating on an old stove**

Zero123++   Instant3D   Ours   Zero123++   Instant3D   Ours

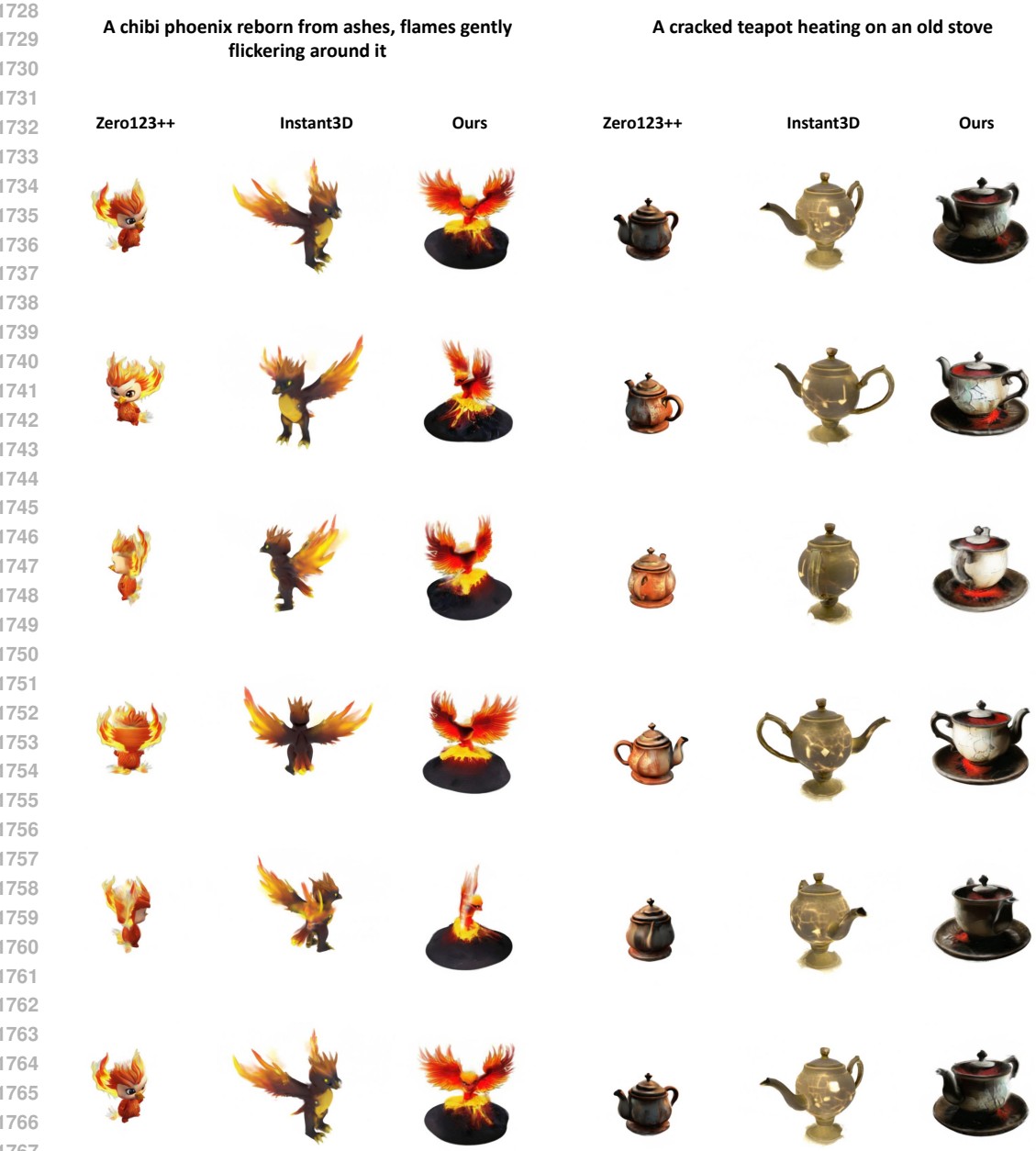

Figure 22: **Visualization of generated objects compared to other edge-cutting methods**

## A.8 MORE RESULTS VISUALIZATION

### A.8.1 COMPARISON WITH OTHER METHODS

### A.8.2 VISUALIZATION OF GENERATED OBJECTS WITH DIFFERENT STYLES

## A.9 BROADER IMPACTS

**Potential positive societal impacts:** The proposed framework, Bootstrap3D, enhances the quality and consistency of 3D models, which can benefit various industries such as entertainment, education, virtual reality, and digital art. By generating and sharing a large synthetic dataset of high-quality synthetic multi-view images, We will promotes open access to resources that can accelerate progress

**A galactic lighthouse guiding traverlers through space-time anomalies**

**A miniature robot companion, poised for adventure with glowing eyes**

Zero123++   Instant3D   Ours   Zero123++   Instant3D   Ours

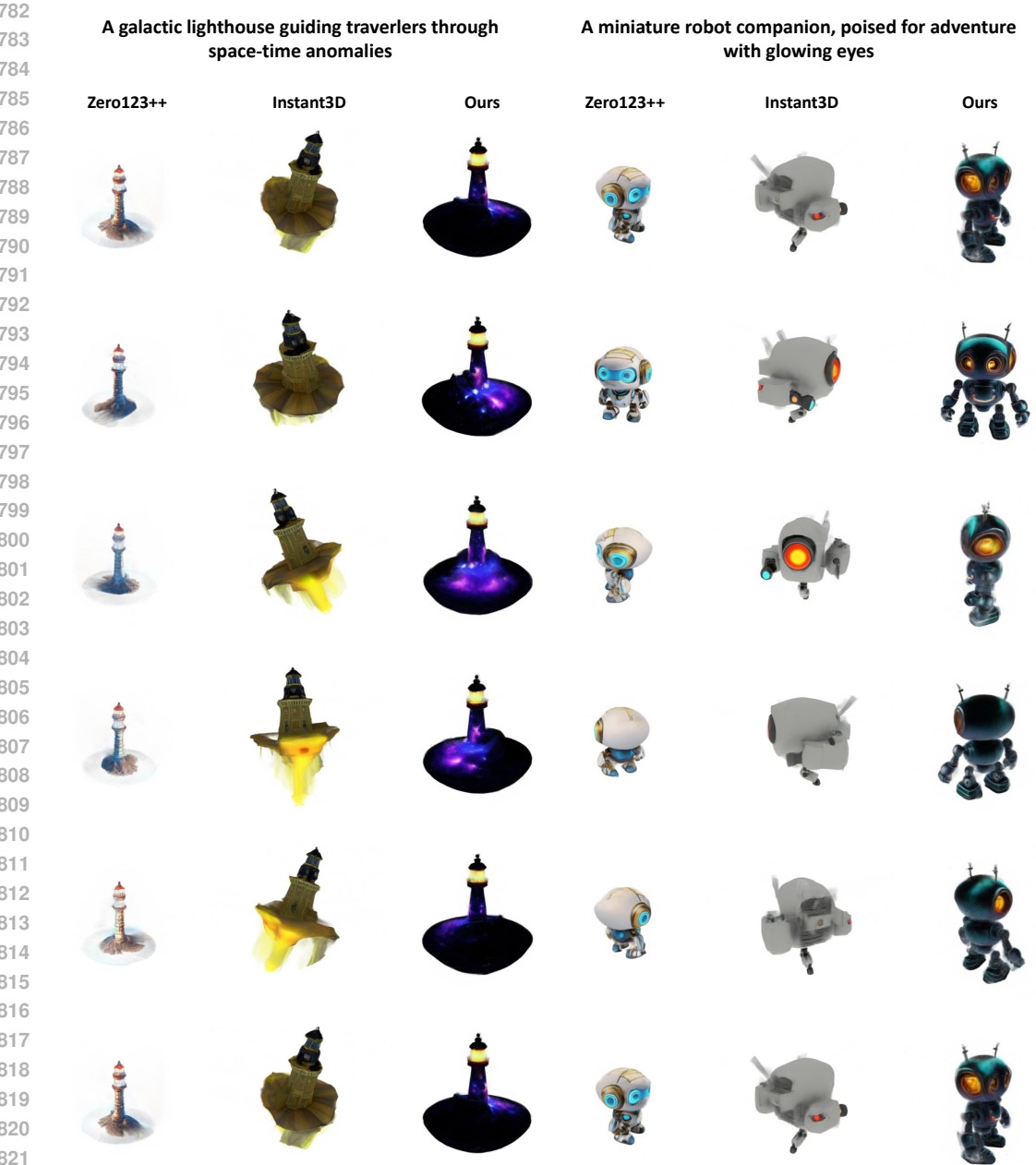

Figure 23: **Visualization of generated objects compared to other edge-cutting methods**

in the field. The model and data can serve as educational tools for students and researchers, fostering learning and innovation in machine learning and 3D modeling.

**Potential negative societal impacts:** High-quality 3D models could be used to create deepfakes or misleading content, which may contribute to disinformation or malicious activities. Monitoring and Defense Mechanisms: Developing tools to detect and prevent the misuse of the generated 3D models, particularly in contexts like disinformation and surveillance. There may be unintended biases in the generated data or models, leading to unfair treatment of specific groups if the technology is deployed in applications affecting societal decision-making.

A tranquil, winter cabin

A serene, celestial observatory

Zero123++    Instant3D    Ours    Zero123++    Instant3D    Ours

Figure 24: **Visualization of generated objects compared to other edge-cutting methods**

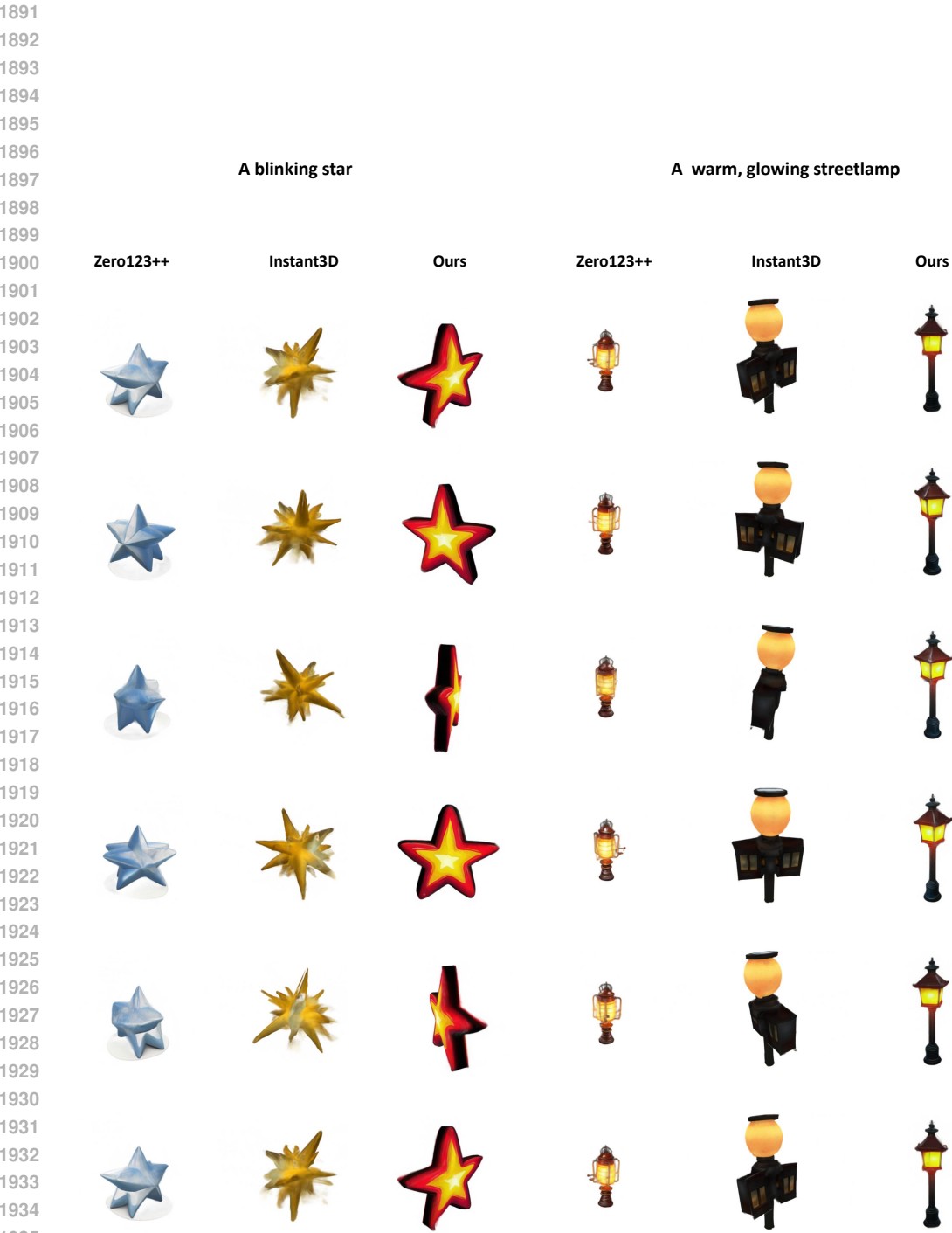

Figure 25: **Visualization of generated objects compared to other edge-cutting methods**

A colossal, stone giant wandering a deserted landscape.

A teal cup, steaming with hot tea.

Zero123++    Instant3D    Ours    Zero123++    Instant3D    Ours

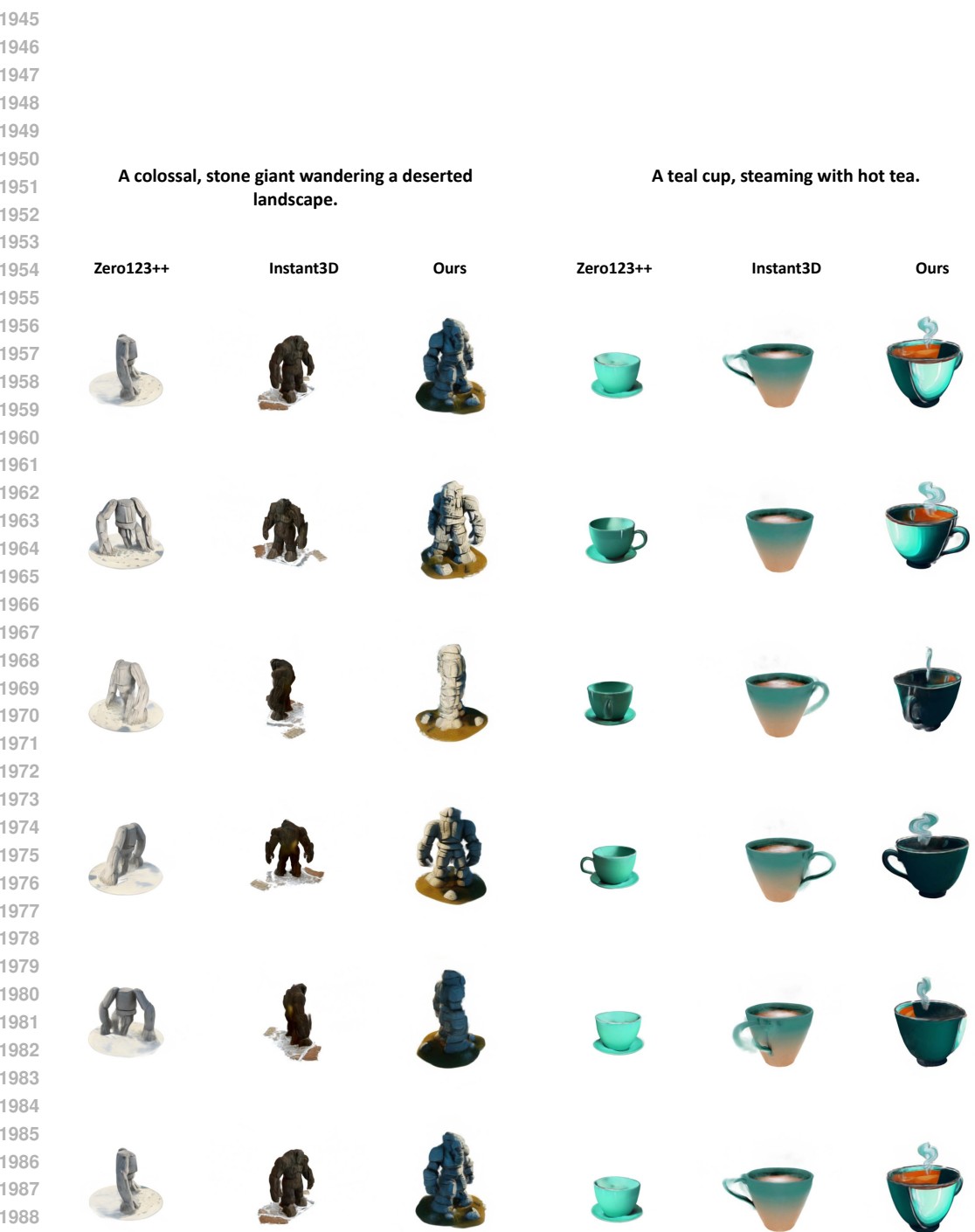

Figure 26: **Visualization of generated objects compared to other edge-cutting methods**

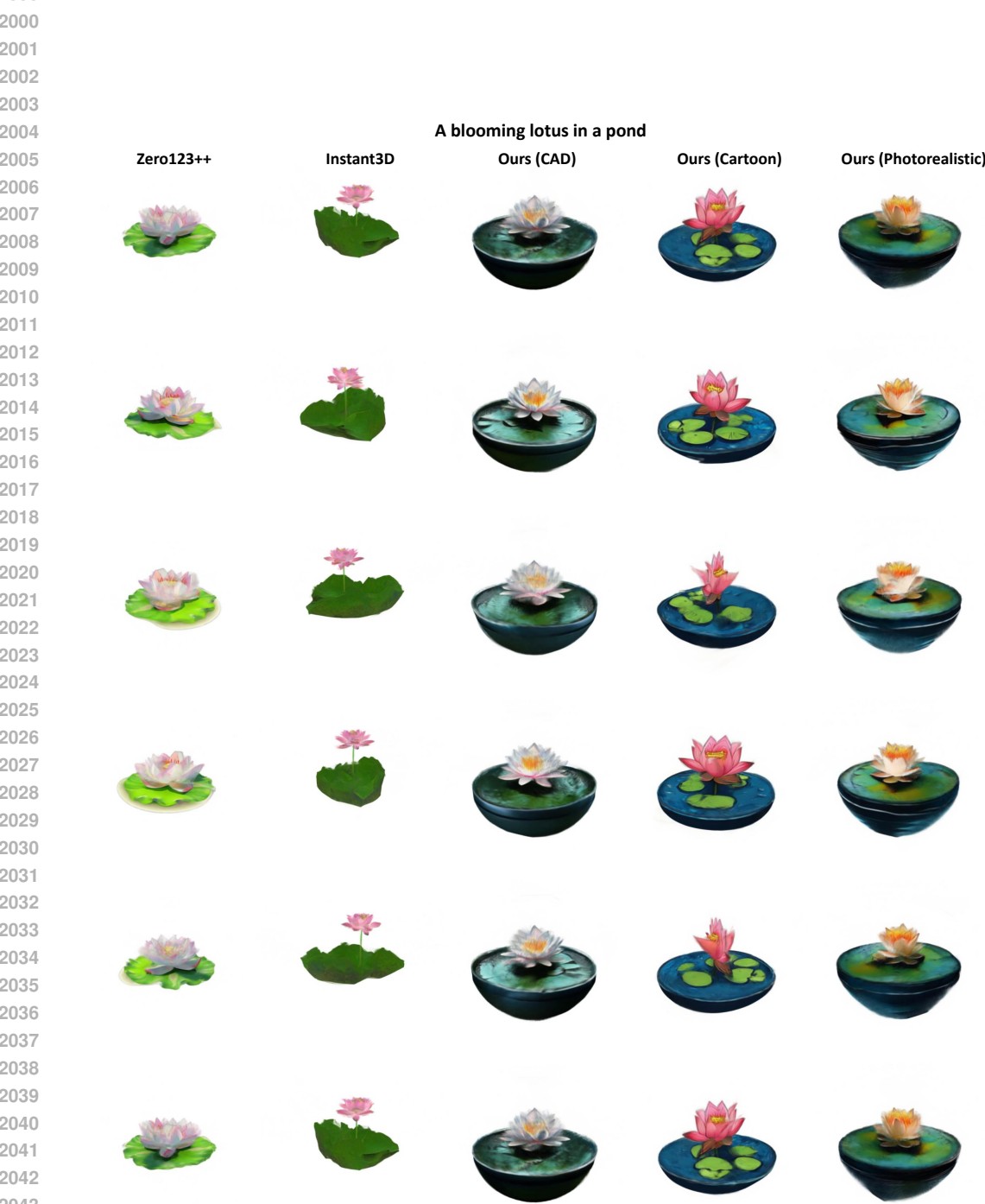

Figure 27: **Visualization of generated objects compared to other edge-cutting methods with different style control.**

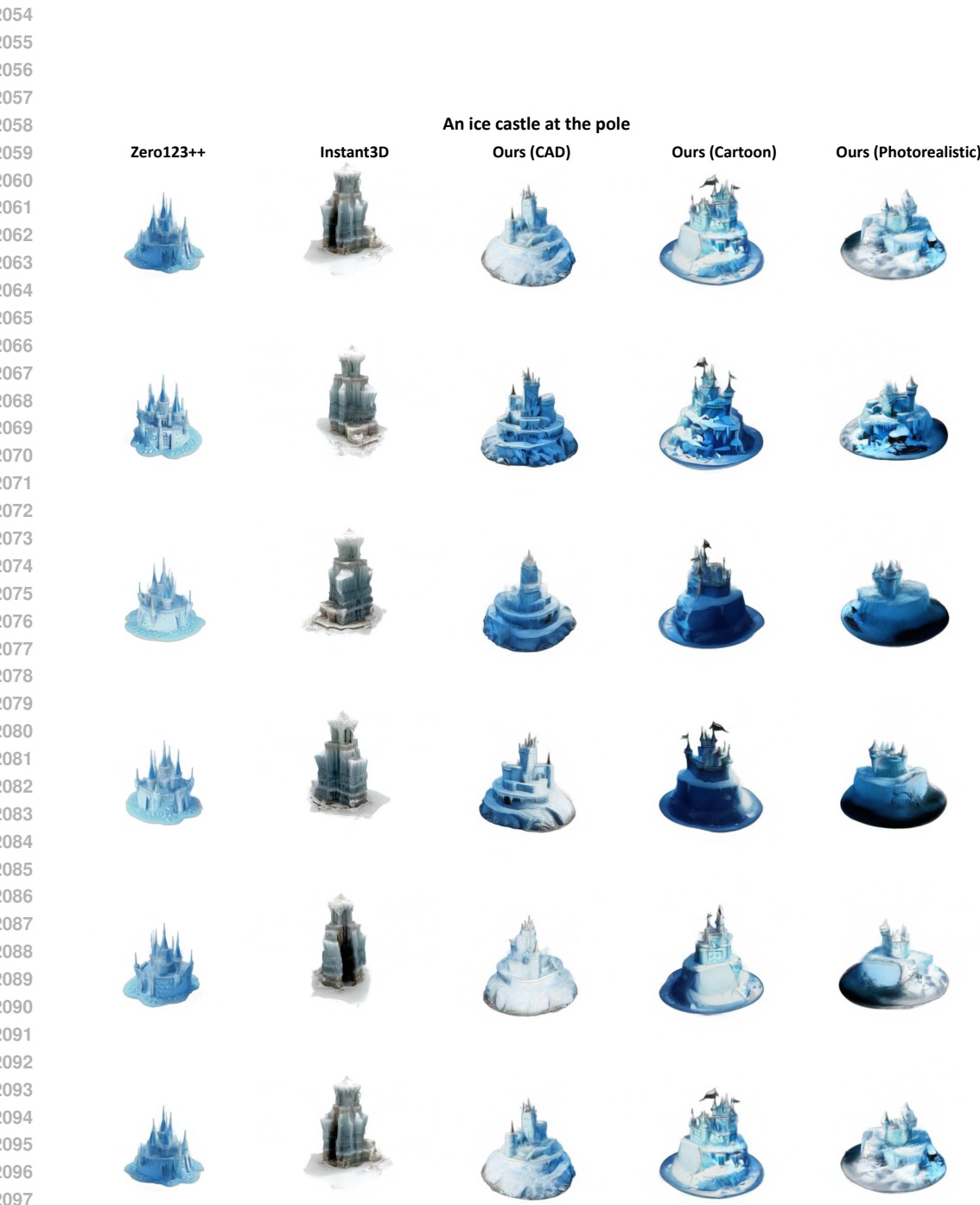

Figure 28: **Visualization of generated objects compared to other edge-cutting methods with different style control.**

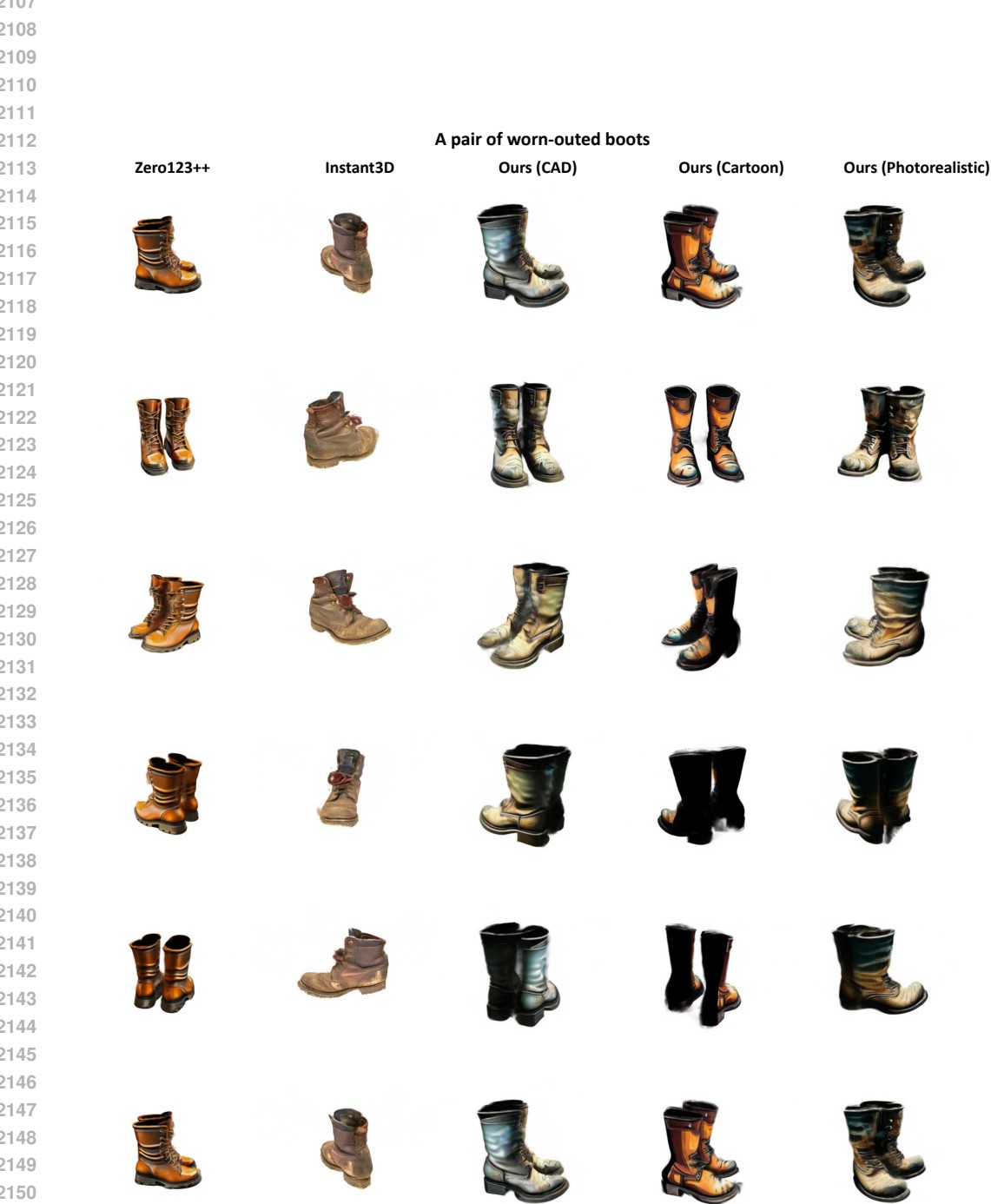

Figure 29: **Visualization of generated objects compared to other edge-cutting methods with different style control.**

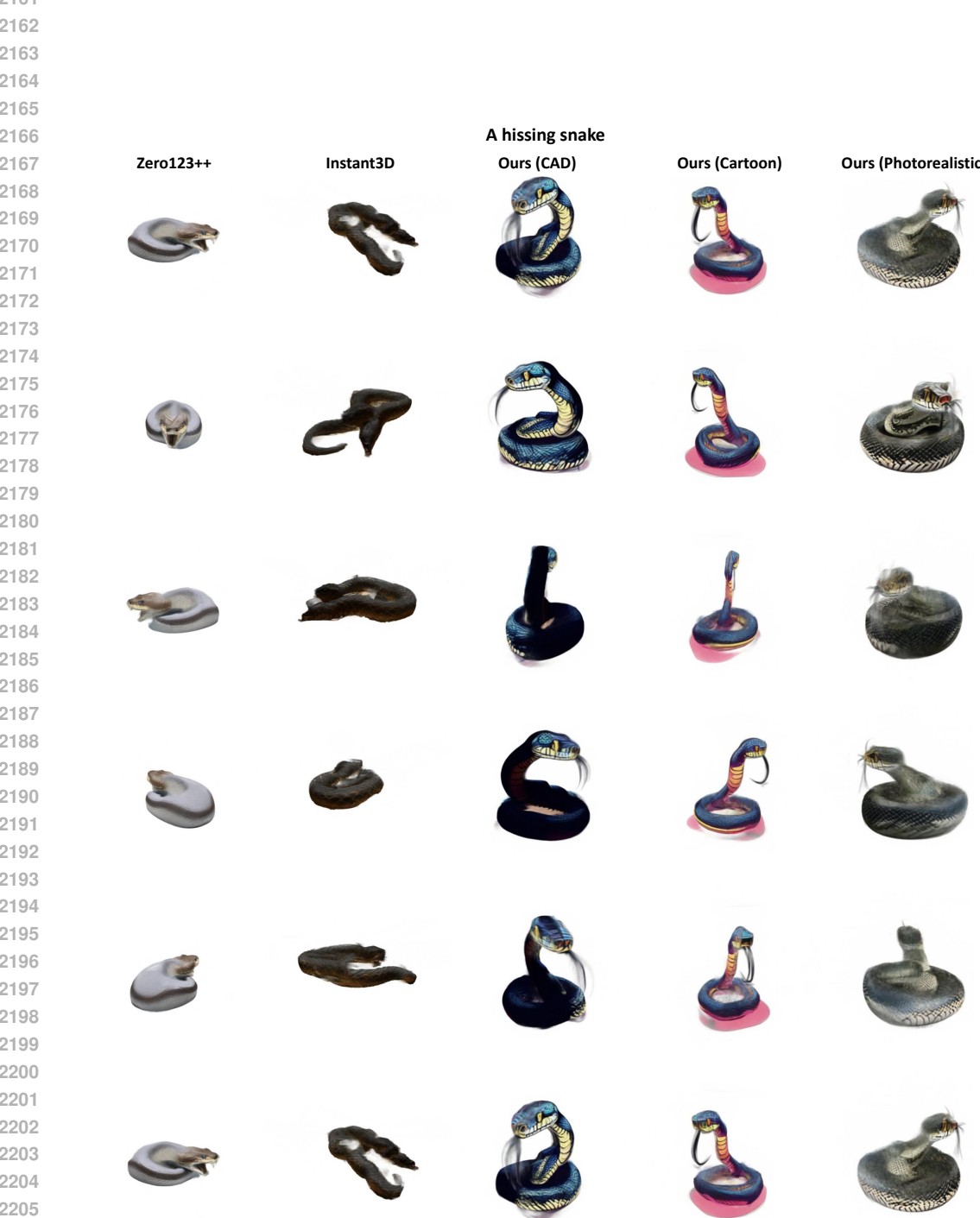

Figure 30: **Visualization of generated objects compared to other edge-cutting methods with different style control.**

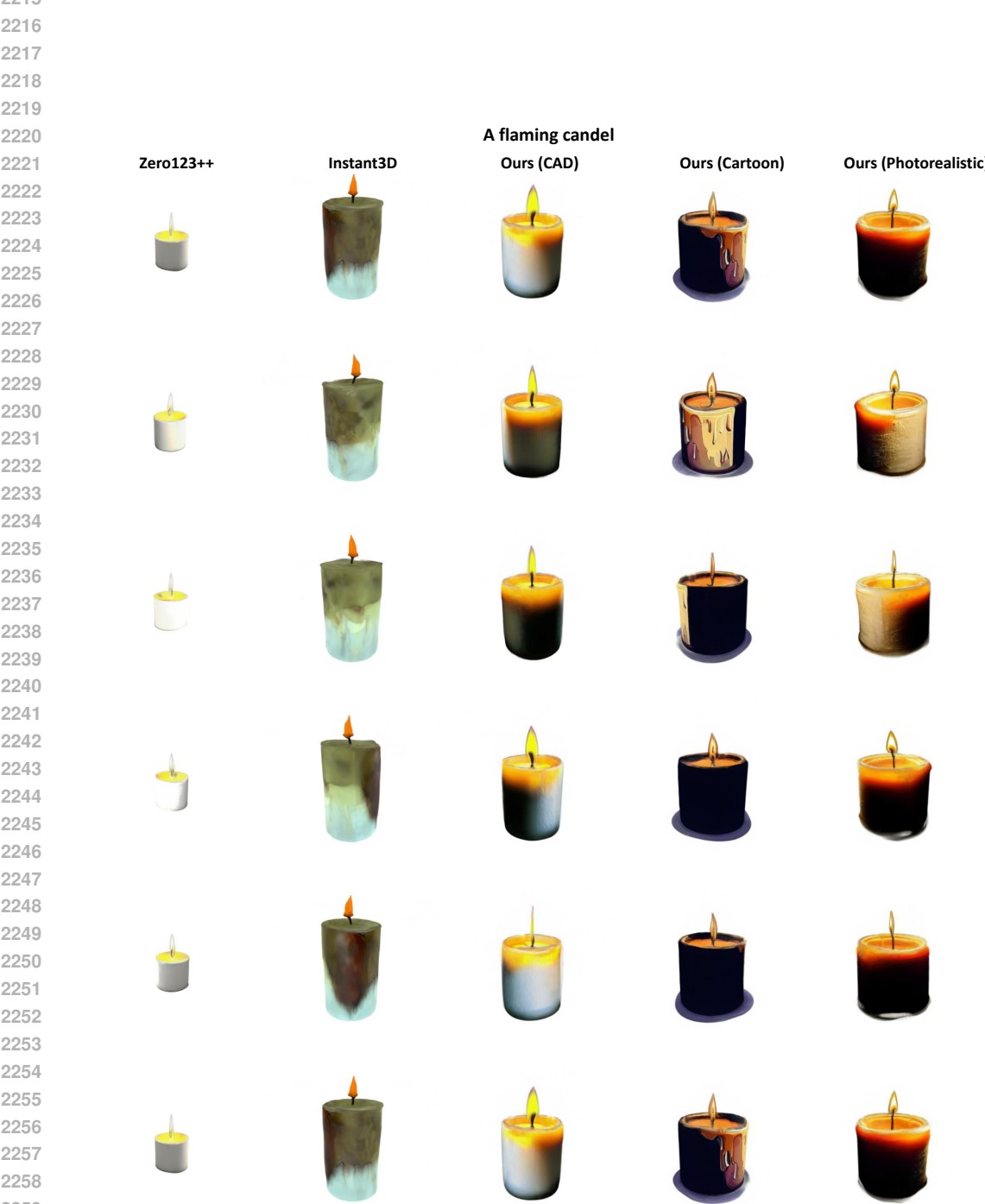

Figure 31: **Visualization of generated objects compared to other edge-cutting methods with different style control.**

