# OpenReview forum: "Bootstrap3D: Improving Multi-view Diffusion Model with Synthetic Data"
_ICLR.cc/2025/Conference — ICLR 2025 Conference Withdrawn Submission_

### Official Review · Reviewer_EhUv · 2024-11-02

**Soundness:** 2
**Presentation:** 2
**Contribution:** 3
**Rating:** 3
**Confidence:** 4

**Summary:**

This paper presents Bootstrap3D, an automated pipeline for multi-view image data generation aimed at enhancing the quality of text-to-multi-view (T2MV) generation. The pipeline consists of three components: (1) Text-to-Image Generation, (2) Image-to-Multi-View Generation, and (3) Quality checks to leave only high-quality data. The authors leverage several advanced tools in their pipeline, including GPT-4, GPT-4V, PixArt-Alpha, SV3D/Zero123++, and GPTeval3D. Additionally, they provide open-source multi-view data assessment tools called MV-LLaVA. They introduce a training timestep reschedule (TTR) to fine-tune PixArt-Alpha with the automatically generated multi-view images and show the effectiveness of their pipeline through CLIP and FID scores on text prompts from GPTeval3D.

**Strengths:**

* The proposed data generation pipeline is practical and addresses significant challenges in the field of 3D generative models.
* The paper clearly describes the challenges of building a data generation pipeline and provides well-documented implementation details that address these challenges.

**Weaknesses:**

**Weakness 1: Clarification about contribution** I think this paper well explains the data generation pipeline, however, additional components such as TTR and MV-LLaVA make it difficult to fully understand the contribution of this paper. Specifically, automated data generation can improve all Text-to-Image-to-Multi-View and Text-to-Multi-View diffusion models combined with TTR.  However, both zero123++ and MVDream used in the comparison were fine-tuned from a variant of stable diffusion, so I don't think it's a fiar comparison. So, the authors should report the comparative results of Stable Diffusion (T2I) and show the performance difference between Zero123++, MVDream, Bootstrap3D fine-tuned on the Stable Diffusion in Table 1 and Table 2.

Also, I am not sure about MV-LLaVA. As I understand (please correct me if I'm wrong), MV-LLaVA is a fine-tuned LLaVA variant to train descriptions and view consistency scores generated by GPT-4V. Is there any specific reason why MV-LLaVA is required for data generation? Can't you just utilize GPT-4V for data generation?

**Weakness 2: Evaluation Metric for 3D reconstruction** This paper only reports CLIP scores and FID scores, which can validate the image-text alignmnets and image quality. Despite the fact that MV-LLaVA provides view consistent synthetic data, the lack of evaluation metrics for 3D reconstruction makes it difficult to verify whether Bootstrap3D's 3D-awareness is actually improved. It would be better to include evaluations of 3D consistency, similar to the COLMAP-based evaluation used in SyncDreamer [1] or the optical flow-based evaluation used in HarmonyView [2].

**Weakness 3: Typo** There are different abbreviations for the same paper. For example, PixArt-Alpa are used in L211 and L249 while PixArt-$\alpha$ are used in L368 and Table 1.

[1] Liu et al., Syncdreamer: Generating multiview-consistent images from a single-view image, ICLR 2024.

[2] Woo et al., Harmonyview: Harmonizing consistency and diversity in one-image-to-3d, CVPR 2024.

**Questions:**

Please see weakness

---

### Official Review · Reviewer_721o · 2024-11-02

**Soundness:** 1
**Presentation:** 2
**Contribution:** 1
**Rating:** 3
**Confidence:** 5

**Summary:**

This paper introduces a data generation pipeline that employs SV3D to generate multi-view images based on elaborated text prompts and fine-tune an MV-LLaVA to select high-quality data and correct inaccurate captions. In addition, they generate 1 million high-quality synthetic multi-view images with dense descriptive captions to deal with the scarcity of high-quality 3D data. They also present a Training Timestep Reschedule (TTR) strategy to guarantee multi-view consistency while maintaining the original 2D diffusion prior.

**Strengths:**

## Stregthness
1. **Motivation is good.** This paper found that the current 3D datasets are low-quality and they research how to generate high-quality datasets for subsequent applications.
2. **Writting is good.** This paper is written well and easy to follow.
3. **High-volume dataset.** This paper generates 1 million multi-view images with dense descriptive captions suitable for training the multi-view diffusion model.

**Weaknesses:**

## Weakness
1. **Purpose Ambiguity.** I understand authors want to generate high-quality multi-view images. However, in line 016, the authors claim that "they propose Bootstrap3D to generate an arbitrary quantity of multi-view images to assist in training multi-view diffusion models". There is ambiguity I am confused. If the proposed Bootstrap3D can generate multi-view images why cannot use it to directly generate an arbitrary number of views of images or reconstruct 3D models while training a multi-view diffusion model? I believe that the authors' research directly using 2D diffusion models or video diffusion models to generate view-consistent images would be more meaningful and useful.

2. **Incorrect Categories.** In lines 039-045, authors identify current methods into two categories: 1. using SDS, 2. 2D diffusion models with multi-view generation. However, there are so many methods in both of these two categories, like MVDream.

3. **Unconvincing Statement.** In lines 047-053, the authors claim that existing datasets, like Objaverse-XL, contain insufficient high-quality data. I don't believe that. The volume of Objaverse-XL is very enormous and this dataset must contain a large number of high-quality data. The authors do not present statistics or visualization. It's very unconvincing.

4. **Dependance on SV3D.** I know that the core multi-view image generation is from SV3D, but why not author research a more robust generation model to generate high-quality data for the target of this paper? It will make this paper outstanding. In addition, why focus on 4 views of generation? SV3D can generate more than 4 views, but in this work, it only can generate 4 views.

5. **Unfair Comparisons.** In Table 1 and Figure 6, the authors do not compare their proposed method with single-image-to-3D methods [1-3] and diffusion-based novel view synthesis methods, like Zero123[4], SyncDreamer[5], and Stable-Zero123[6]. These methods can also generate multi-view images, meaningful to compare. More importantly, there is no user study. It's very important to measure the quality of generation qualitatively by real users.

   [1] Chen A, Xu H, Esposito S, et al. Lara: Efficient large-baseline radiance fields[J]. arXiv preprint arXiv:2407.04699, 2024.

   [2] Xu Y, Tan H, Luan F, et al. Dmv3d: Denoising multi-view diffusion using 3d large reconstruction model[J]. arXiv preprint arXiv:2311.09217, 2023.

   [3] Sun J, Zhang B, Shao R, et al. Dreamcraft3d: Hierarchical 3d generation with bootstrapped diffusion prior[J]. arXiv preprint arXiv:2310.16818, 2023.

   [4] Liu R, Wu R, Van Hoorick B, et al. Zero-1-to-3: Zero-shot one image to 3d object[C]//Proceedings of the IEEE/CVF international conference on computer vision. 2023: 9298-9309.

   [5] Liu Y, Lin C, Zeng Z, et al. Syncdreamer: Generating multiview-consistent images from a single-view image[J]. arXiv preprint arXiv:2309.03453, 2023.

   [6] https://stability.ai/stable-3d



6. **Unconvincing Experiments.** The purpose of this paper is to generate a sufficient number of multi-view images for training diffusion models. However, I don't find any experiments about the authors using their generated datasets to train a large 3D generation model and compare it in 3D format.


7. **Something Confused.** I am still confused why this work cannot be directly used for 3D generation but the purpose is to assist in other diffusion models. Also, this work relies on SV3D to generate multi-view images, but why the results can be better than SV3D? Whether authors' SV3D fintuned or not? Is there any post-processing?



8. **Limited Novelty.** In summary, I think the purpose of this paper is to generate any number of multi-view images, like dataset curation.
However, the novelty is quite limited. In addition, this work is a little bit engineering and I think this work is more suitable for the NIPS dataset track.

**Questions:**

1. Why does this work only generate images with 4 views but not more?
2. Why does this work rely on SV3D to generate images but better than this model?
3. Why not this work can be directly used for SDS and subsequent 3D generation?
4. Inference time is very important.

---

### Official Review · Reviewer_Hgjo · 2024-11-03

**Soundness:** 2
**Presentation:** 3
**Contribution:** 2
**Rating:** 3
**Confidence:** 4

**Summary:**

The paper proposes a new method for training multi-view diffusion models. The main idea is to add synthetic data obtained by existing conditional multi-view generation models to the training data. The paper proposes a pipeline for obtaining high quality synthetic data that involves prompt generation with LLMs, single and multi view image generation, and quality check with a finetuned large multimodal model.

**Strengths:**

1. The idea of bootstrapping data from existing multi-view generation models to train a new multi-view generation is novel and interesting. This is similar to the pseudo-labeling technique in semi-supervised learning.

2. The qualitative results shown in the paper are good compared to other existing methods.

3. The presentation of the pipeline is detailed and clear with helpful illustration and diagrams.

**Weaknesses:**

1. In section 4 the paper compares the proposed method to existing methods like Instant3D and MVDream, but the diffusion backbone of these methods are different. The proposed method uses PixArt-Alpha, while Instant3D and MVDream use SDXL. It is unclear whether the authors re-implement Instant3D and MVDream with the PixArt-Alpha backbone. The comparison is unfair if we use different backbones, because we do not know if the performance gain is due to a better training scheme or just a more powerful backbone. For example, the Instant3D paper shows that we can get significantly better results by simply switching from SD1.5 to SDXL.

2. In the proposed data generation pipeline, the quality check stage seems to be the most important. Otherwise we do not need to train a new model, but can just use the previous stages of the pipeline to be the generator in inference time (single-view -> mulit-view with existing model). But I do not find any ablation for the quality check component. The closest I can find is Table 3, but it only provides ablation results for the entire data generation process but not individual components.

3. Qualitative comparison is much more important than quantitative comparison for this kind of models. It would be better to have a large number of uncurated random samples from different models to compare.

4. I cannot find qualitative comparison between a naive model trained without synthetic data and the proposed full model.

**Questions:**

1. How much do you think the PixArt backbone contributes to the performance gain over other methods using SDXL, compared to the gain obtained by using synthetic data?

2. I suggest doing more careful ablation study for your pipeline, especially for the quality check component.

3. I suggest providing more uncurated qualitative results, as suggested in the weaknesses section.

---

### Official Review · Reviewer_36JU · 2024-11-04

**Soundness:** 3
**Presentation:** 3
**Contribution:** 3
**Rating:** 6
**Confidence:** 4

**Summary:**

This paper aims to improving the quality of text to multi-view diffusion model by using synthetic data. They use text-to-image model to generate single image, then use single-tp-multiview generator (SV3D, zero123++) to generate multiveiw images.

To filter the inconsistent images, they fine-tune the LLaVA model specifically for multi-view  assessment.

They also propose to use Training Timestep Reschedule (TTR) to balance the usage of synthetic data.

There are both quantitative and qualitative evaluations.

**Strengths:**

1. The motivation is clear that the current limitation for multi-view model , compared to single image model, is the data. Using existing text-to-multiview model to generate more data, and use them for training make sense.
2. The way of filtering mv data is important. As the synthetic data might not be multiview consistent, getting a good set of data is important. The idea of train a MV-LLaVA is interesting.
3. The use of synthetic data is also important. Even after filtering, the data is still not prefect. By using synthetic for large t training and small t for real data, the model could benefit from larger data while maintaining high quality from real data.
4. The metric shows the  bootstrapped model surpasses the data generator (SV3D, zero123++).

**Weaknesses:**

1. Lack of multi-view consistency evaluation. Although there is result using LRM etc, direct measuring of view consistency is not explicit. One way is to use LRM to get a 3D asset, then use reprojection error.

**Questions:**

1. See weakness.
2. Given model such as LRM,  GRM ,why not use them for synthetic data generation?
3. Will the bootstrap approach be applicable for true 3D generation as well?

---

### Note · Authors · 2024-11-15

I have read and agree with the venue's withdrawal policy on behalf of myself and my co-authors.